# DOUBLE GENERATIVE ADVERSARIAL NETWORKS FOR CONDITIONAL INDEPENDENCE TESTING

## ABSTRACT

In this article, we consider the problem of high-dimensional conditional independence testing, which is a key building block in statistics and machine learning. We propose a double generative adversarial networks (GANs)-based inference procedure. We first introduce a double GANs framework to learn two generators, and integrate the two generators to construct a doubly-robust test statistic. We next consider multiple generalized covariance measures, and take their maximum as our test statistic. Finally, we obtain the empirical distribution of our test statistic through multiplier bootstrap. We show that our test controls type-I error, while the power approaches one asymptotically. More importantly, these theoretical guarantees are obtained under much weaker and practically more feasible conditions compared to existing tests. We demonstrate the efficacy of our test through both synthetic and real data examples.

## 1 INTRODUCTION

Conditional independence (CI) is a fundamental concept in statistics and machine learning. Testing conditional independence is a key building block and plays a central role in a wide variety of statistical learning problems, for instance, causal inference (Pearl, 2009), graphical models (Koller & Friedman, 2009), dimension reduction (Li, 2018), among others. In this article, we aim at testing whether two random variables $X$ and $Y$ are conditionally independent given a set of confounding variables $Z$. That is, we test the hypotheses:

$$\mathcal{H}_0 : X \perp\!\!\!\perp Y \mid Z \qquad \text{versus} \qquad \mathcal{H}_1 : X \not\perp\!\!\!\perp Y \mid Z, \tag{1}$$

given the observed data of $n$ i.i.d. copies $\{(X_i, Y_i, Z_i)\}_{1 \leq i \leq n}$ of $(X, Y, Z)$. For our problem, $X, Y$ and $Z$ can all be multivariate. However, the main challenge arises when the confounding set of variables $Z$ is high-dimensional. As such, we primarily focus on the scenario with a univariate $X$ and $Y$, and a multivariate $Z$. Meanwhile, our proposed method can be extended to the multivariate $X$ and $Y$ scenario as well. Another challenge is the limited sample size compared to the dimensionality of $Z$. As a result, many existing tests are ineffective, with either an inflated type-I error, or not having enough power to detect the alternatives. See Section 2 for a detailed review.

We propose a double generative adversarial networks (GANs, Goodfellow et al., 2014)-based inference procedure for the CI testing problem (1). Our proposal involves two key components, a double GANs framework to learn two generators that approximate the conditional distribution of $X$ given $Z$ and $Y$ given $Z$, and a maximum of generalized covariance measures of multiple combinations of the transformation functions of $X$ and $Y$. We first establish that our test statistic is doubly-robust, which offers additional protections against potential misspecification of the conditional distributions (see Theorems 1 and 2). Second, we show the resulting test achieves a valid control of the type-I error asymptotically, and more importantly, under the conditions that are much weaker and practically more feasible (see Theorem 3). Finally, we prove the power of our test approaches one asymptotically (see Theorem 4), and demonstrate it is more powerful than the competing tests empirically.

## 2 RELATED WORKS

There has been a growing literature on conditional independence testing in recent years; see (Li & Fan, 2019) for a review. Broadly speaking, the existing testing methods can be cast into four main categories, the metric-based tests, e.g., (Su & White, 2007; 2014; Wang et al., 2015), the conditional randomization-based tests (Candes et al., 2018; Bellot & van der Schaar, 2019), the kernel-based

tests (Fukumizu et al., 2008; Zhang et al., 2011), and the regression-based tests (Hoyer et al., 2009; Zhang et al., 2018; Shah & Peters, 2018). There are other types of tests, e.g., Bergsma (2004); Doran et al. (2014); Sen et al. (2017; 2018); Berrett et al. (2019), to mention a few.

The metric-based tests typically employ some kernel smoothers to estimate the conditional characteristic function or the distribution function of $Y$ given $X$ and $Z$. Kernel smoothers, however, are known to suffer from the curse of dimensionality, and as such, these tests are not suitable when the dimension of $Z$ is high. The conditional randomization-based tests require the knowledge of the conditional distribution of $X|Z$ (Candes et al., 2018). If unknown, the type-I error rates of these tests rely critically on the quality of the approximation of this conditional distribution. Kernel-based test is built upon the notion of maximum mean discrepancy (MMD, Gretton et al., 2012), and could have inflated type-I errors. The regression-based tests have valid type-I error control, but may suffer from inadequate power. Next, we discuss in detail the conditional randomization-based tests, in particular, the work of Bellot & van der Schaar (2019), the regression-based and the MMD-based tests, since our proposal is closely related to them.

## 2.1 CONDITIONAL RANDOMIZATION-BASED TESTS

The family of conditional randomization-based tests is built upon the following basis. If the conditional distribution $P_{X|Z}$ of $X$ given $Z$ is known, then one can independently draw $X_i^{(1)} \sim P_{X|Z=Z_i}$ for $i = 1, \ldots, n$, and these samples are independent of the observed samples $X_i$'s and $Y_i$'s. Write $\boldsymbol{X} = (X_1, \ldots, X_n)^\top$, $\boldsymbol{X}^{(1)} = (X_1^{(1)}, \ldots, X_n^{(1)})^\top$, $\boldsymbol{Y} = (Y_1, \ldots, Y_n)^\top$, and $\boldsymbol{Z} = (Z_1, \ldots, Z_n)^\top$. Here we use boldface letters to denote data matrices that consist of $n$ samples. The joint distributions of $(\boldsymbol{X}, \boldsymbol{Y}, \boldsymbol{Z})$ and $(\boldsymbol{X}^{(1)}, \boldsymbol{Y}, \boldsymbol{Z})$ are the same under $\mathcal{H}_0$. Any large difference between the two distributions can be interpreted as the evidence against $\mathcal{H}_0$. Therefore, one can repeat the process $M$ times, and generate $X_i^{(m)} \sim P_{X|Z=Z_i}$, $i = 1, \ldots, n$, $m = 1, \ldots, M$. Write $\boldsymbol{X}^{(m)} = (X_1^{(m)}, \ldots, X_n^{(m)})^\top$. Then, for any given test statistic $\rho = \rho(\boldsymbol{X}, \boldsymbol{Y}, \boldsymbol{Z})$, its associated $p$-value is $p = \left[1 + \sum_{m=1}^M \mathbb{I}\{\rho(\boldsymbol{X}^{(m)}, \boldsymbol{Y}, \boldsymbol{Z}) \geq \rho(\boldsymbol{X}, \boldsymbol{Y}, \boldsymbol{Z})\}\right]/(1 + M)$, where $\mathbb{I}(\cdot)$ is the indicator function. Since the triplets $(\boldsymbol{X}, \boldsymbol{Y}, \boldsymbol{Z}), (\boldsymbol{X}^{(1)}, \boldsymbol{Y}, \boldsymbol{Z}), \ldots, (\boldsymbol{X}^{(M)}, \boldsymbol{Y}, \boldsymbol{Z})$ are exchangeable under $\mathcal{H}_0$, the $p$-value is valid, and it satisfies that $\Pr(p \leq \alpha | \mathcal{H}_0) \leq \alpha + o(1)$ for any $0 < \alpha < 1$.

In practice, however, $P_{X|Z}$ is rarely known, and Bellot & van der Schaar (2019) proposed to approximate it using GANs. Specifically, they learned a generator $\mathbb{G}_X(\cdot, \cdot)$ from the observed data, then took $Z_i$ and a noise variable $v_{i,X}^{(m)}$ as input to obtain a sample $\widetilde{X}_i^{(m)}$, which minimizes the divergence between the distributions of $(X_i, Z_i)$ and $(\widetilde{X}_i^{(m)}, Z_i)$. The $p$-value is then computed by replacing $\boldsymbol{X}^{(m)}$ by $\widetilde{\boldsymbol{X}}^{(m)} = (\widetilde{X}_1^{(m)}, \ldots, \widetilde{X}_n^{(m)})^\top$. They called this test GCIT, short for generative conditional independence test. By Theorem 1 of Bellot & van der Schaar (2019), the excess type-I error of this test is upper bounded by

$$\Pr(p \leq \alpha | \mathcal{H}_0) - \alpha \leq \mathrm{E} d_{\mathrm{TV}}(\widetilde{P}_{\boldsymbol{X}|\boldsymbol{Z}}, P_{\boldsymbol{X}|\boldsymbol{Z}}) = \mathrm{E} \sup_A |\Pr(\boldsymbol{X} \in A | \boldsymbol{Z}) - \Pr(\widetilde{\boldsymbol{X}}^{(m)} \in A | \boldsymbol{Z})| \equiv D, \quad (2)$$

where $d_{\mathrm{TV}}$ is the total variation norm between two probability distributions, the supremum is taken over all measurable sets, and the expectations in (2) are taken with respect to $\boldsymbol{Z}$.

By definition, the quantity $D$ on the right-hand-side of (2) measures the quality of the conditional distribution approximation. Bellot & van der Schaar (2019) argued that this error term is negligible due to the capacity of deep neural nets in estimating conditional distributions. To the contrary, we find this approximation error is usually *not* negligible, and consequently, it may inflate the type-I error and invalidate the test. We consider a simple example to further elaborate this.

**Example 1.** Suppose $X$ is one-dimensional, and follows a simple linear regression model, $X = Z^\top \beta_0 + \varepsilon$, where the error $\varepsilon$ is independent of $Z$ and $\varepsilon \sim N(0, \sigma_0^2)$ for some $\sigma_0^2 > 0$.

Suppose we know a priori that the linear regression model holds. We thus estimate $\beta_0$ by ordinary least squares, and denote the resulting estimator by $\widehat{\beta}$. For simplicity, suppose $\sigma_0^2$ is known too. For this simple example, we have the following result regarding the approximation error term $D$.

**Proposition 1** *Suppose the linear regression model holds. The derived distribution $\widetilde{P}_{\boldsymbol{X}|\boldsymbol{Z}}$ is $N(\boldsymbol{Z}\widehat{\beta}, \sigma_0^2 I_n)$, where $I_n$ is the $n \times n$ identity matrix. Then $D$ is not $o(1)$.*

To facilitate the understanding of the convergence behavior of $D$, we sketch a few lines of the proof of Proposition 1. A detailed proof is given in Appendix F.1. Let $\widetilde{P}_{X|Z=Z_i}$ denote the conditional distribution of $\widetilde{X}_i^{(m)}$ given $Z_i$, which is $N(Z_i^\top \widehat{\beta}, \sigma_0^2)$ in this example. If $D = o(1)$, then,

$$\widetilde{D} \equiv n^{1/2}\sqrt{\mathrm{E}d_{\mathrm{TV}}^2(\widetilde{P}_{X|Z=Z_i}, P_{X|Z=Z_i})} = o(1). \tag{3}$$

In other words, the validity of GCIT requires the root mean squared total variation distance in (3) to converge at a faster rate than $n^{-1/2}$. However, this rate cannot be achieved in general. In our simple Example 1, we have $\widetilde{D} \geq c$ for some universal constant $c > 0$. Consequently, $D$ in (2) is not $o(1)$. Proposition 1 shows that, even if we know a priori that the linear model holds, $D$ is not to decay to zero as $n$ grows to infinity. In practice, we do not have such prior model information. Then it would be even more difficult to estimate the conditional distribution $P_{X|Z}$. Therefore, using GANs to approximate $P_{X|Z}$ does not guarantee a negligible approximation error, nor the validity of the test.

## 2.2 REGRESSION-BASED TESTS

The family of regression-based tests is built upon a key quantity, the generalized covariance measure,

$$\mathrm{GCM}(X, Y) = \frac{1}{n}\sum_{i=1}^n \left\{X_i - \widehat{\mathrm{E}}(X_i|Z_i)\right\}\left\{Y_i - \widehat{\mathrm{E}}(Y_i|Z_i)\right\},$$

where $\widehat{\mathrm{E}}(X|Z)$ and $\widehat{\mathrm{E}}(Y|Z)$ are the predicted condition mean $\mathrm{E}(X|Z)$ and $\mathrm{E}(Y|Z)$, respectively, by any supervised learner. When the prediction errors of $\widehat{\mathrm{E}}(X|Z)$ and $\widehat{\mathrm{E}}(Y|Z)$ satisfy certain convergence rates, Shah & Peters (2018) proved that GCM is asymptotically normal. Under $\mathcal{H}_0$, the asymptotic mean of GCM is zero, and its asymptotic standard deviation can be consistently estimated by some standard error estimator, denoted by $\widehat{s}(\mathrm{GCM})$. Therefore, at level $\alpha$, we reject $\mathcal{H}_0$, if $|\mathrm{GCM}|/\widehat{s}(\mathrm{GCM})$ exceeds the upper $\alpha/2$th quantile of a standard normal distribution.

Such a test is valid. However, it may not have sufficient power to detect $\mathcal{H}_1$. This is because the asymptotic mean of GCM equals $\mathrm{GCM}^*(X, Y) = \mathrm{E}\{X - \mathrm{E}(X|Z)\}\{Y - \mathrm{E}(Y|Z)\}$. The regression-based tests require $|\mathrm{GCM}^*|$ to be nonzero under $\mathcal{H}_1$ to have power. However, there is no guarantee of this requirement. We again consider a simple example to elaborate.

**Example 2.** Suppose $X^*$, $Y$ and $Z$ are independent random variables. Besides, $X^*$ has mean zero, and $X = X^*g(Y)$ for some function $g$.

For this example, we have $\mathrm{E}(X|Z) = \mathrm{E}(X)$, since both $X^*$ and $Y$ are independent of $Z$, and so is $X$. Besides, $\mathrm{E}(X) = \mathrm{E}(X^*)\mathrm{E}\{g(Y)\} = 0$, since $X^*$ is independent of $Y$ and $\mathrm{E}(X^*) = 0$. As such, $\mathrm{GCM}^*(X, Y) = \mathrm{E}\{X - \mathrm{E}(X)\}\{Y - \mathrm{E}(Y|Z)\} = 0$ for any function $g$. On the other hand, $X$ and $Y$ are conditionally dependent given $Z$, as long as $g$ is not a constant function. Therefore, for this example, the regression-based tests would fail to discriminate between $\mathcal{H}_0$ and $\mathcal{H}_1$.

## 2.3 MMD-BASED TESTS

The family of kernel-based tests often involves the notion of maximum mean discrepancy as a measure of independence. For any two probability measures $P, Q$ and a function space $\mathbb{F}$, define

$$\mathrm{MMD}(P, Q|\mathbb{F}) = \sup_{f\in\mathbb{F}}\left\{\mathrm{E}f(W_1) - \mathrm{E}f(W_2)\right\}, \quad W_1 \sim P,\ W_2 \sim Q.$$

Let $\mathbb{H}_1, \mathbb{H}_2$ be some function spaces of $X$ and $Y$. Define $\phi_{XY} = \mathrm{MMD}(P_{XY}, Q_{XY}|\mathbb{H}_1 \otimes \mathbb{H}_2)$, where $\otimes$ is the tensor product, $P_{XY}$ is the joint distribution of $(X, Y)$, and $Q_{XY}$ is the conditionally independent distribution with the same $X$ and $Y$ margins as $P_{XY}$. Then following the calculations in Appendix D, we have, $\phi_{XY} = \sup_{h_1\in\mathbb{H}_1, h_2\in\mathbb{H}_2}\mathrm{E}[h_1(X) - \mathrm{E}\{h_1(X)|Z\}][h_2(Y) - \mathrm{E}\{h_2(Y)|Z\}]$. We see that $\phi_{XY}$ measures the average conditional association between $X$ and $Y$ given $Z$. Under $\mathcal{H}_0$, it equals zero, and hence an estimator of this measure can be used as a test statistic for $\mathcal{H}_0$.

## 3 A NEW DOUBLE GANs-BASED TESTING PROCEDURE

We propose a double GANs-based testing procedure for the conditional independence testing problem (1). Conceptually, our test integrates GCIT, regression-based and MMD-based tests. Meanwhile, our new test addresses the limitations of the existing ones. Unlike GCIT that only learned the

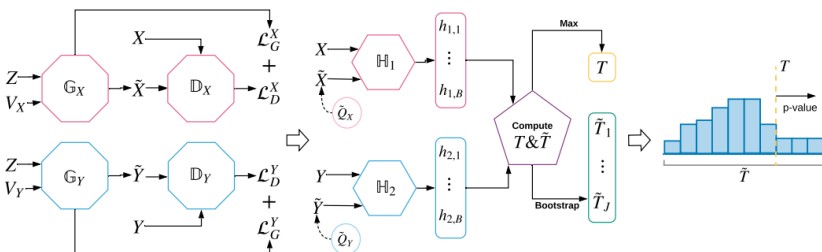

Figure 1: Illustration of the conditional independence test with double GANs.

conditional distribution of $X|Z$, we learn two generators $\mathbb{G}_X$ and $\mathbb{G}_Y$ to approximate the conditional distributions of both $X|Z$ and $Y|Z$. We then integrate the two generators in an appropriate way to construct a doubly-robust test statistic, and we only require the root mean squared total variation norm to converge at a rate of $n^{-\kappa}$ for some $\kappa > 1/4$. Such a requirement is much weaker and practically more feasible than the condition in (3). The notion of doubly-robustness property comes from the classical semiparametric theory in statistics (Tsiatis, 2007). Specifically, a doubly robust procedure applies both types of models simultaneously, and produces a consistent estimateif either of the two models has been consistently estimated.

Moreover, to improve the power of the test, we consider a set of the GCMs, $\{\text{GCM}(h_1(X), h_2(Y)) : h_1, h_2\}$, for multiple combinations of transformation functions $h_1(X)$ and $h_2(Y)$. We then take the maximum of all these GCMs as our test statistic. This essentially yields $\phi_{XY}$, which is connected with the notion of MMD. To see why the maximum-type statistic can enhance the power, we quickly revisit Example 2. When $g$ is not a constant function, there exists some nonlinear function $h_1$ such that $h_1^*(Y) = \text{E}\{h_1(X)|Y\}$ is not a constant function of $Y$. Set $h_2 = h_1^*$. We have $\text{GCM}^* = \text{E}[h_1\{X^*g(Y)\}\{Y - \text{E}(Y)\}] = \text{Var}\{h_1^*(Y)\} > 0$. This enables us to discriminate $\mathcal{H}_1$ from $\mathcal{H}_0$.

We next detail our testing procedure. A graphical overview is given in Figure 1.

### 3.1 TEST STATISTIC

We begin with two function spaces, $\mathbb{H}_1 = \{h_{1,\theta_1} : \theta_1 \in \mathbb{R}^{d_1}\}$ and $\mathbb{H}_2 = \{h_{2,\theta_2} : \theta_2 \in \mathbb{R}^{d_2}\}$, indexed by some parameters $\theta_1$ and $\theta_2$, respectively. In our implementation, we set $\mathbb{H}_1$ and $\mathbb{H}_2$ to the classes of neural networks with a single-hidden layer, finitely many hidden nodes, and the sigmoid activation function. We then randomly generate $B$ functions, $h_{1,1}, \ldots, h_{1,B} \in \mathbb{H}_1$, $h_{2,1}, \ldots, h_{2,B} \in \mathbb{H}_2$, where we independently generate i.i.d. multivariate normal variables $\theta_{1,1}, \ldots, \theta_{1,B} \sim N(0, 2I_{d_1}/d_1)$, and $\theta_{2,1}, \ldots, \theta_{2,B} \sim N(0, 2I_{d_2}/d_2)$. We then set $h_{1,b} = h_{1,\theta_{1,b}}$, and $h_{2,b} = h_{2,\theta_{2,b}}$, $b = 1, \ldots, B$. Consider the following maximum-type test statistic,

$$\max_{b_1, b_2 \in \{1, \ldots, B\}} \widehat{\sigma}_{b_1, b_2}^{-1} \left| \frac{1}{n} \sum_{i=1}^{n} \left[ h_{1,b_1}(X_i) - \widehat{\text{E}}\{h_{1,b_1}(X_i)|Z_i\} \right] \left[ h_{2,b_2}(Y_i) - \widehat{\text{E}}\{h_{2,b_2}(Y_i)|Z_i\} \right] \right|, \quad (4)$$

where $\widehat{\sigma}_{b_1,b_2}^2$ is a consistent estimator for $\sqrt{n}\text{GCM}(h_1(X), h_2(Y))$. See Section A for definition. To compute (4), however, we need to estimate the conditional means $\text{E}\{h_{1,b_1}(X)|Z\}$, $\text{E}\{h_{2,b_2}(Y)|Z\}$ for $b_1, b_2 = 1, \ldots, B$. In theory, $B$ should diverge to infinity to guarantee the power property of the test. Separately applying supervised learning algorithms $2B$ times to compute these means is computationally very expensive. Instead, we propose to implement this step based on the generators $\mathbb{G}_X$ and $\mathbb{G}_Y$ estimated using GANs, which is computationally much more efficient.

Specifically, for $i = 1, \ldots, n$, we randomly generate i.i.d. random vectors $\{v_{i,X}^{(m)}\}_{m=1}^{M}$, $\{v_{i,Y}^{(m)}\}_{m=1}^{M}$ and output the pseudo samples $\widetilde{X}_i^{(m)} = \mathbb{G}_X(Z_i, v_{i,X}^{(m)})$, $\widetilde{Y}_i^{(m)} = \mathbb{G}_Y(Z_i, v_{i,Y}^{(m)})$, for $m = 1, \ldots, M$, to approximate the conditional distributions of $X_i$ and $Y_i$ given $Z_i$. We then compute $\widehat{\text{E}}\{h_{1,b_1}(\widetilde{X}_i)|Z_i\} = M^{-1}\sum_{m=1}^{M} h_{1,b_1}(X_i^{(m)})$, and $\widehat{\text{E}}\{h_{2,b_2}(Y_i)|Z_i\} = M^{-1}\sum_{m=1}^{M} h_{2,b_2}(\widetilde{Y}_i^{(m)})$, for $b_1, b_2 = 1, \ldots, B$. Plugging those estimated means into (4) produces our test statistic, $T \equiv \max_{b_1, b_2} \left| n^{-1/2} \sum_{i=1}^{n} \psi_{b_1, b_2, i} \right|$, where

$$\psi_{b_1, b_2, i} = \widehat{\sigma}_{b_1,b_2}^{-1} \left\{ h_{1,b_1}(X_i) - \frac{1}{M} \sum_{m=1}^{M} h_{1,b_1}\left( \widetilde{X}_i^{(m)} \right) \right\} \left\{ h_{2,b_2}(Y_i) - \frac{1}{M} \sum_{m=1}^{M} h_{2,b_2}\left( \widetilde{Y}_i^{(m)} \right) \right\}.$$

---

**Input:** Number of functions $B$, number of pseudo samples $M$, and number of data splits $L$.

**Step 1:** Divide $\{1, \ldots, n\}$ into $L$ folds $\mathcal{I}^{(1)}, \ldots, \mathcal{I}^{(L)}$. Let $\mathcal{I}^{(-\ell)} = \{1, \ldots, n\} - \mathcal{I}^{(\ell)}$.

**Step 2:** For $\ell = 1, \ldots, L$, train two generators $\mathbb{G}_X^{(\ell)}$ and $\mathbb{G}_Y^{(\ell)}$ based on $\{(X_i, Z_i)\}_{i \in \mathcal{I}^{(-\ell)}}$ and $\{(Y_i, Z_i)\}_{i \in \mathcal{I}^{(-\ell)}}$, to approximate the conditional distributions of $X|Z$ and $Y|Z$.

**Step 3:** For $\ell = 1, \ldots, L$ and $i \in \mathcal{I}_\ell$, generate i.i.d. random noises $\left\{v_{i,X}^{(m)}\right\}_{m=1}^M$, $\left\{v_{i,Y}^{(m)}\right\}_{m=1}^M$. Set $\widetilde{X}_i^{(m)} = \mathbb{G}_X^{(\ell)}\left(Z_i, v_{i,X}^{(m)}\right)$, and $\widetilde{Y}_i^{(m)} = \mathbb{G}_Y^{(\ell)}\left(Z_i, v_{i,Y}^{(m)}\right)$, $m = 1, \ldots, M$.

**Step 4:** Randomly generate $h_{1,1}, \ldots, h_{1,B} \in \mathbb{H}_1$ and $h_{2,1}, \ldots, h_{2,B} \in \mathbb{H}_2$.

**Step 5:** Compute the test statistic $T$.

---

**Algorithm 1:** Algorithm for computing the test statistic.

To help reduce the type-I error of our test, we further employ a data splitting and cross-fitting strategy, which is commonly used in statistical testing (Romano & DiCiccio, 2019). That is, we use different subsets of data samples to learn GANs and to construct the test statistic. We summarize our procedure of computing the test statistic in Algorithm 1.

### 3.2 BOOTSTRAPPING THE $p$-VALUE

Next, we propose a multiplier bootstrap method to approximate the distribution of $\sqrt{n}T$ under $\mathcal{H}_0$ to compute the corresponding $p$-value. The key observation is that $\psi_{b_1, b_2} = n^{-1/2} \sum_{i=1}^n \psi_{b_1, b_2, i}$ is asymptotically normal with zero mean under $\mathcal{H}_0$; see the proof of Theorem 3 in Appendix F.3 for details. As such, $\sqrt{n}T = \max_{b_1, b_2} |n^{-1/2} \sum_{i=1}^n \psi_{b_1, b_2, i}|$ is to converge to a maximum of normal variables in absolute values.

To approximate this limiting distribution, we first estimate the covariance matrix of a $B^2$-dimensional vector formed by $\{\psi_{b_1, b_2}\}_{b_1, b_2}$ using the sample covariance matrix $\widehat{\Sigma}$. We then generate i.i.d. random vectors with the covariance matrix equal to $\widehat{\Sigma}$, and compute the maximum elements of each of these vectors in absolute values. Finally, we use these maximum absolute values to approximate the distribution of $T$ under the null. We summarize this procedure in Algorithm 2.

### 3.3 APPROXIMATING CONDITIONAL DISTRIBUTION VIA GANS

We adopt the proposal in Genevay et al. (2017) to learn the conditional distributions $P_{X|Z}$ and $P_{Y|Z}$. Recall that $\widetilde{P}_{X|Z}$ is the distribution of pseudo outcome generated by the generator $\mathbb{G}_X$ given $Z$. We consider estimating $P_{X|Z}$ by optimizing $\min_{\mathbb{G}_X} \max_c \widetilde{\mathcal{D}}_{c,\epsilon}(P_{X|Z}, \widetilde{P}_{X|Z})$, where $\widetilde{\mathcal{D}}_{c,\epsilon}$ denotes the Sinkhorn loss function between two probability measures with respect to some cost function $c$ and some regularization parameter $\epsilon > 0$. A detailed definition of $\widetilde{\mathcal{D}}_{c,\epsilon}$ is given in Appendix B. Intuitively, the closer the two probability measures, the smaller the Sinkhorn loss. As such, maximizing the loss with respect to the cost function learns a discriminator that can better discriminate the samples generated between $P_{X|Z}$ and $\widetilde{P}_{X|Z}$. On the other hand, minimizing the maximum cost with respect to the generator $\mathbb{G}_X$ makes it closer to the true distribution $P_{X|Z}$. This yields the minimax

---

**Input:** Number of bootstrap samples $J$, and $\{\psi_{b_1, b_2, i}\}_{b_1, b_2, i}$.

**Step 1:** Compute a $B^2 \times B^2$ matrix $\widehat{\Sigma}$ whose $\{b_1 + B(b_2 - 1), b_3 + B(b_4 - 1)\}$th entry is given by $(n-1)^{-1} \sum_{i=1}^n (\psi_{b_1, b_2, i} - \psi_{b_1, b_2})(\psi_{b_3, b_4, i} - \psi_{b_3, b_4})$.

**Step 2:** Generate i.i.d. standard normal variables $Z_{j,b}$ for $j = 1, \ldots, J$, $b = 1, \ldots, B^2$. Set $\mathbf{Z}_j = (Z_{j,1}, \ldots, Z_{j,B^2})^\top$, and $\widetilde{T}_j = \|\widehat{\Sigma}^{1/2} \mathbf{Z}_j\|_\infty$, where $\widehat{\Sigma}^{1/2}$ is a positive semi-definite matrix that satisfies $\widehat{\Sigma}^{1/2} \widehat{\Sigma}^{1/2} = \widehat{\Sigma}$, and $\| \cdot \|_\infty$ is the maximum element of a vector in absolute values.

**Step 3:** Compute the $p$-value, $p = J^{-1} \sum_{j=1}^J \mathbb{I}(T \geq \widetilde{T}_j)$.

---

**Algorithm 2:** Algorithm for computing the $p$-value.

formulation $\min_{\mathbb{G}_X} \max_c \widetilde{\mathcal{D}}_{c,\epsilon}(P_{X|Z}, \widetilde{P}_{X|Z})$ that we target. In practice, we approximate the cost and the generator based on neural networks. Integrations in the objective function $\widetilde{\mathcal{D}}_{c,\epsilon}(P_{X|Z}, \widetilde{P}_{X|Z})$ are approximated by sample averages. A pseudocode detailing our learning procedure is given in Appendix B. The conditional distribution $P_{Y|Z}$ is estimated similarly.

We make two remarks. First, our proposed framework is general, and any GANs learning procedure can be applied. In our implementation, we have chosen the Sinkhorn GANs because of its competitive performance. We did not use the original GANs due to its instability and the mode collapse issue. We did implement WGANs, but found the estimated distributions may suffer from the large bias resulting from the gradient penalty enforced in WGANs. Second, it is important to check the "goodness-of-fit" of the generator. In practice, this could be achieved by comparing the conditional histogram of the generated samples to that of the true samples. See Appendix C for details.

## 4 ASYMPTOTIC THEORY

To derive the theoretical properties of the test statistic $T$, we first introduce a concept of the "oracle" test statistic $T^*$. If $P_{X|Z}$ and $P_{Y|Z}$ were known a priori, then one can draw $\{X_i^{(m)}\}_m$ and $\{Y_i^{(m)}\}_m$ from $P_{X|Z=Z_i}$ and $P_{Y|Z=Z_i}$ directly, and can compute the test statistic $T$ by replacing $\{\widetilde{X}_i^{(m)}\}_m$ and $\{\widetilde{Y}_i^{(m)}\}_m$ with $\{X_i^{(m)}\}_m$ and $\{Y_i^{(m)}\}_m$. We call the resulting $T^*$ an "oracle" test statistic.

We next establish the double-robustness property of $T$, which helps us better understand why our proposed test can relax the requirement in (3). Informally speaking, the double-robustness means that $T$ is asymptotically equivalent to $T^*$ when either the conditional distribution of $X|Z$, or that of $Y|Z$ is well approximated by GANs. It guarantees $T$ converges to $T^*$ at a faster rate than the estimated conditional distribution. In contrast, the convergence rate of the GCIT test statistic is the same as the estimated conditional distribution. As such, our procedure requires a weaker condition.

**Theorem 1 (Double-robustness)** *Suppose $M$ is proportional to $n$, and $B = c_0 n^c$ for some constants $c, c_0 > 0$. Then $T - T^* = o_p(1)$, when either $\left( E[d_{TV}^2\{\widetilde{Q}_X^{(\ell)}(\cdot|Z), Q_X(\cdot|Z)\}]\right)^{1/2} = o(\log^{-1/2} n)$, or $\left( E[d_{TV}^2\{\widetilde{Q}_Y^{(\ell)}(\cdot|Z), Q_Y(\cdot|Z)\}]\right)^{1/2} = o(\log^{-1/2} n)$.*

The conditions on $M$ and $B$ are mild, as these are user-specified parameters. As we have discussed, when both total variation distances converge to zero, the test statistic $T$ converges at a faster rate than those total variation distances. Therefore, we can greatly relax the condition in (3), and replace it with, for any $\ell = 1, \ldots, L$,

$$\left[E\{d_{\text{TV}}^2(\widetilde{P}_{X|Z}^{(\ell)}, P_{X|Z})\}\right]^{1/2} = O(n^{-\kappa}), \quad \text{and} \quad \left[E\{d_{\text{TV}}^2(\widetilde{P}_{Y|Z}^{(\ell)}, P_{Y|Z})\}\right]^{1/2} = O(n^{-\kappa}), \tag{5}$$

for some constant $0 < \kappa < 1/2$, where $\widetilde{P}_{X|Z}^{(\ell)}$ and $\widetilde{P}_{Y|Z}^{(\ell)}$ denote the conditional distributions approximated via GANs trained on the $\ell$-th subset. The next theorem summarizes this discussion.

**Theorem 2** *Suppose the conditions in Theorem 1 and 5 holds. Then $T - T^* = O_p(n^{-2\kappa} \log n)$.*

Since $\kappa > 0$, the convergence rate of $(T - T^*)$ is faster than that in (5). To ensure $\sqrt{n}(T - T^*) = o_p(1)$, it suffices to require $\kappa > 1/4$. In contrast to (3), this rate is achievable. We consider three examples to illustrate, while the condition holds in a wide range of settings.

**Example 3 (Parametric setting).** Suppose the parametric forms of $Q_X$ and $Q_Y$ are correctly specified. Then the requirement $\kappa > 1/4$ holds if $k = O(n^{t_0})$ for some $t_0 < 1/4$, where $k$ is the dimension of the parameters defining the parametric model.

**Example 4. (Nonparametric setting with binary data).** Suppose $X, Y$ are binary variables. Then it suffices to estimate the conditional means of $X$ and $Y$ given $Z$. The requirement $\kappa > 1/4$ holds if the mean squared prediction errors of both nonparametric estimators are $O(n^{\kappa_0})$ for some $\kappa_0 > 1/4$.

**Example 5. (Nonparametric setting with general data).** Suppose $X, Y$ are continuous variables. We apply GANs to learn the condition distributions. Chen et al. (2020) established the statistical properties of GANs, and one can apply their technical tools to check the convergence rates in (5). In general, the convergence rate depends on the smoothness of the conditional density function. The smoother the conditional density, the faster the rate.

Next, we establish the size and the power properties of our proposed test.

**Theorem 3** *Suppose the conditions in Theorem 1 hold. Suppose (5) holds for some $\kappa > 1/4$. Then the p-value from Algorithm 2 satisfies that $Pr(p \leq \alpha | \mathcal{H}_0) = \alpha + o(1)$.*

Theorem 3 shows that our proposed test can control the type-I error. Next, to derive the power property, we introduce the pair of hypotheses based on the notion of weak CI (Daudin, 1980):

$$\mathcal{H}_0^* : \text{ECov}(f(X), g(Y)|Z) = 0, \forall f \in L_X^2, g \in L_Y^2 \text{ versus } \mathcal{H}_1^* : \text{ECov}(f(X), g(Y)|Z) \neq 0, \exists f, g,$$

where $L_X^2$ and $L_Y^2$ denote the class of all squared integrable functions of $X$ and $Y$, respectively. We note that conditional independence implies weak conditional independence, i.e., $\mathcal{H}_0$ implies $\mathcal{H}_0^*$. Consequently, $\mathcal{H}_1^*$ implies $\mathcal{H}_1$. The next theorem shows that our proposed test is consistent against the alternatives in $\mathcal{H}_1^*$, but not against all alternatives in $\mathcal{H}_1$.

**Theorem 4** *Suppose the conditions in Theorem 3 hold. Then the p-value from Algorithm 2 satisfies that $Pr(p \leq \alpha | \mathcal{H}_1^*) \to 1$, as $n \to \infty$.*

Finally, we remark that our test is constructed based on $\phi_{XY}$. Meanwhile, we may consider another test based on $\phi_{XYZ} = \text{MMD}(P_{XYZ}, Q_{XYZ} | \mathbb{H}_1 \otimes \mathbb{H}_2 \otimes \mathbb{H}_3)$, where $P_{XYZ}$ is the joint distribution of $(X, Y, Z)$, $Q_{XYZ} = P_{X|Z} P_{Y|Z} P_Z$, and $\mathbb{H}_3$ is a neural network class of functions of $Z$. This type of test is consistent against all alternatives in $\mathcal{H}_1$. However, in our numerical experiments, we find it less powerful compared to our test. This agrees with the observation by Li & Fan (2019) in that, even though the tests based on weak CI cannot fully characterize CI, they potentially benefit from an improved power.

## 5 NUMERICAL STUDIES

We give the implementation details of our testing procedure in Appendix A. The time complexity of our test is dominated by Step 2 of Algorithm 1, where we use GANs to estimate the conditional distributions $P_{X|Z}$ and $P_{Y|Z}$. The complexity of each SGD iteration is $O(RN^2)$; see Appendix B. All experiments were run on 16 N1 CPUs on Google Cloud Computing platform. The wall clock time for computing a single test statistic was about 2.5 minutes.

### 5.1 SYNTHETIC DATA EXAMPLE

We generate synthetic data following the post non-linear noise model similarly as in Zhang et al. (2011); Doran et al. (2014); Bellot & van der Schaar (2019),

$$X = \sin(a_f^\top Z + \varepsilon_f) \quad \text{and} \quad Y = \cos(a_g^\top Z + bX + \varepsilon_g).$$

The entries of $a_f, a_g$ are randomly and uniformly sampled from $[0, 1]$, then normalized to unit norm. The noise variables $\varepsilon$ are independently sampled from a normal distribution with mean zero

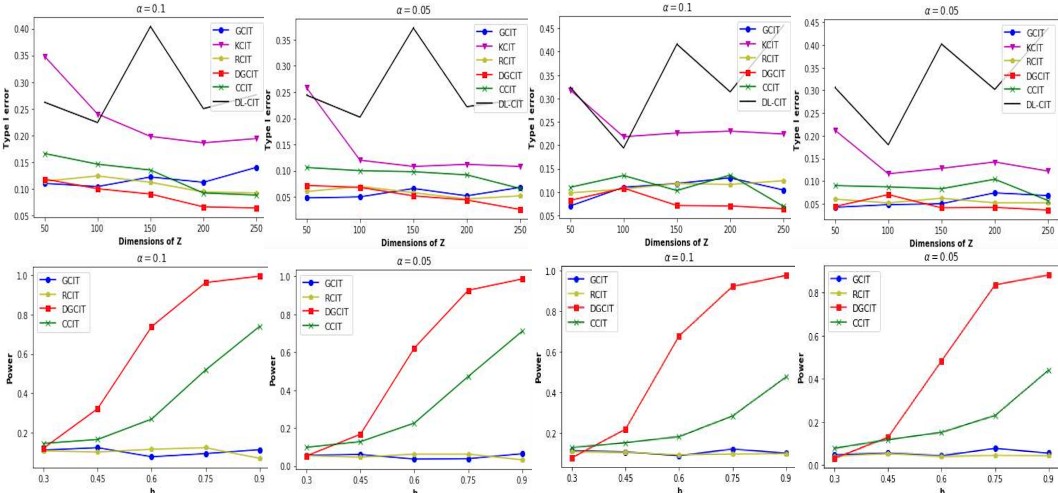

Figure 2: Top panels: the empirical type-I error rate of various tests under $\mathcal{H}_0$. From left to right: normal $Z$ with $\alpha = 0.1$, normal $Z$ with $\alpha = 0.05$, Laplacian $Z$ with $\alpha = 0.1$, and Laplacian $Z$ with $\alpha = 0.05$. Bottom panels: the empirical power of various tests under $\mathcal{H}_1$. From left to right: $d_Z = 100, \alpha = 0.1$, $d_Z = 100, \alpha = 0.05$, $d_Z = 200, \alpha = 0.1$, and $d_Z = 200, \alpha = 0.05$.

Table 1: The variable importance measures of the elastic net and random forest models, versus the $p$-values of the GCIT and DGCIT tests for the anti-cancer drug example.

|  | BRAF.V600E | BRAF.MC | HIP1 | FTL3 | CDC42BPA | THBS3 | DNMT1 | PRKD1 | PIP5K1A | MAP3K5 |
|---|---|---|---|---|---|---|---|---|---|---|
| EN | 1 | 3 | 4 | 5 | 7 | 8 | 9 | 10 | 19 | 78 |
| RF | 1 | 2 | 3 | 14 | 8 | 34 | 28 | 18 | 7 | 9 |
| GCIT | <0.001 | <0.001 | 0.008 | 0.521 | 0.050 | 0.013 | 0.020 | 0.002 | 0.001 | <0.001 |
| DGCIT | 0 | 0 | 0 | 0 | 0 | 0 | 0 | 0 | 0 | 0.794 |

and variance $0.25$. In this model, the parameter $b$ determines the degree of conditional dependence. When $b = 0$, $\mathcal{H}_0$ holds, and otherwise $\mathcal{H}_1$ holds. The sample size is fixed at $n = 1000$.

We call our test DGCIT, short for double GANs-based conditional independence test. We compare it with the GCIT test of Bellot & van der Schaar (2019), the regression-based test (RCIT) of Shah & Peters (2018), the kernel MMD-based test (KCIT) of Zhang et al. (2011), the classifier CI test (CCIT) of Sen et al. (2017) and the deep learning-based CI test (DL-CIT) of Sen et al. (2018).

**Type-I error under** $\mathcal{H}_0$. We vary the dimension of $Z$ as $d_Z = 50, 100, 150, 200, 250$, and consider two generation distributions. We first generate $Z$ from a standard normal distribution, then from a Laplace distribution. We set the significance level at $\alpha = 0.05$ and $0.1$. Figure 2 top panels report the empirical size of the tests aggregated over 500 data replications. We make the following observations. First, the type-I error rates of our test and RCIT are close to or below the nominal level in nearly all cases. Second, KCIT and DL-CIT fail in that their type-I errors are considerably larger than the nominal level in all cases. Third, GCIT and CCIT have inflated type-I errors in some cases. Take GCIT as an example. When $Z$ is normal, $d_Z = 250$ and $\alpha = 0.1$, its empirical size is close to $0.15$. This is consistent with our discussion in Section 2.1, as GCIT requires a very strong condition to control the type-I error.

**Powers under** $\mathcal{H}_1$. We generate $Z$ from a standard normal distribution, with $d_Z = 100, 200$, and vary the value of $b = 0.3, 0.45, 0.6, 0.75, 0.9$ that controls the magnitude of the alternative. Figure 2 bottom panels report the empirical power of the tests over 500 data replications. We observe that our test is the most powerful, and the empirical power approaches 1 as $b$ increases to $0.9$, demonstrating the consistency of the test. Meanwhile, both GCIT and RCIT have no power in all cases. We did not report the power of KCIT and DL-CIT, because as we show earlier, they can not control the size, and thus they empirical powers are meaningless.

## 5.2 ANTI-CANCER DRUG DATA EXAMPLE

We illustrate our proposed test with an anti-cancer drug dataset from the Cancer Cell Line Encyclopedia (Barretina et al., 2012). We concentrate on a subset, the CCLE data, that measures the treatment response of drug PLX4720. It is well known that the patient's cancer treatment response to drug can be strongly influenced by alterations in the genome (Garnett et al., 2012) This data measures 1638 genetic mutations of $n = 472$ cell lines, and the goal of our analysis is to determine which genetic mutation is significantly correlated with the drug response after conditioning on all other mutations. The same data was also analyzed in Tansey et al. (2018) and Bellot & van der Schaar (2019). We adopt the same screening procedure as theirs to screen out irrelevant mutations, which leaves a total of 466 potential mutations for our conditional independence testing.

The ground truth is unknown for this data. Instead, we compare with the variable importance measures obtained from fitting an elastic net (EN) model and a random forest (RF) model as reported in Barretina et al. (2012). In addition, we compare with the GCIT test of Bellot & van der Schaar (2019). Table 1 reports the corresponding variable importance measures and the $p$-values, for 10 mutations that were also reported by Bellot & van der Schaar (2019). We see that, the $p$-values of the tests generally agree well with the variable important measures from the EN and RF models. Meanwhile, the two conditional independence tests agree relatively well, except for two genetic mutations, MAP3K5 and FTL3. GCIT concluded that MAP3K5 is significant ($p < 0.001$) but FTL3 is not ($p = 0.521$), whereas our test leads to the opposite conclusion that MAP3K5 is insignificant ($p = 0.794$) but FTL3 is ($p = 0$). Besides, both EN and RF place FTL3 as an important mutation. We then compare our findings with the cancer drug response literature. Actually, MAP3K5 has not been previously reported in the literature as being directly linked to the PLX4720 drug response. Meanwhile, there is strong evidence showing the connections of the FLT3 mutation with cancer response (Tsai et al., 2008; Larrosa-Garcia & Baer, 2017). Combining the existing literature with our theoretical and synthetic results, we have more confidence about the findings of our proposed test.

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

## A    ADDITIONAL DETAILS ABOUT THE PROPOSED TESTING PROCEDURE

We first discuss the computation time. In our synthetic data example, it took about 2.5 minutes to compute our test, 2 minutes to compute CCIT, and 20 seconds to compute GCIT and DL-CIT.

We next discuss some implementation details. For the number of functions $B$ in Algorithm 1, it represents a trade-off. By Theorem 4, $B$ should be as large as possible to guarantee a good power. In practice, the computation complexity increases as $B$ increases. Our numerical studies suggest that the value of $B$ between 30 and 50 achieves a good balance between the power and the computational cost, and we fix $B = 30$. For the number of pseudo samples $M$, and the number of sample splittings $L$, we find the results are not overly sensitive to their choices, and thus we fix $M = 100$ and $L = 3$. For the GANs, we use a single-hidden layer neural network to approximate both the discriminator and generator. The number of nodes in the hidden layer is set at 128. The dimension of the input noise $\nu_{i,X}^{(m)}$ and $\nu_{i,Y}^{(m)}$ is set at 10. The performance of GANs is largely affected by the regularization parameter $\epsilon$ and the number of Sinkhorn iterations $R$. In our experiments, we set $\epsilon = 0.8$ and $R = 30$. In practice, we suggest to tune these parameters by investigating the goodness-of-fit of the resulting generator. This could be achieved by comparing the conditional histogram of the generated samples to that of the true samples. See Appendix C for details. We use stochastic gradient descent (SGD) to update the parameters in GANs. The batch size $N$ is set to 64. We find the resulting GANs work reasonably well in our experiments.

Finally, we introduce the variance estimator. Define

$$\widehat{\sigma}_{b_1,b_2}^2 = \max\left\{ \frac{1}{n-1}\sum_{i=1}^{n}\left( \left[h_{1,b_1}(X_i) - \widehat{\mathrm{E}}\{h_{1,b_1}(X_i)|Z_i\}\right]\left[h_{2,b_2}(Y_i) - \widehat{\mathrm{E}}\{h_{2,b_2}(Y_i)|Z_i\}\right] \right.\right.$$
$$\left.\left. -\mathrm{GCM}\{h_{1,b_1}(X), h_{2,b_2}(Y)\} \right)^2, \epsilon_0 \right\},$$

for any $b_1$ and $b_2$, where $\epsilon_0$ denotes some sufficiently small constant. The constant $\epsilon_0$ is to guarantee that the denominator of $\psi_{b_1,b_2,i}$ is strictly greater than zero, such that the proposed test has the desired size and power properties.

## B    ADDITIONAL DETAILS FOR CONDITIONAL DISTRIBUTION APPROXIMATION USING GANS

We first introduce the notion of optimal transport (OT). Let $\mathcal{X}$ and $\mathcal{Y}$ be some closed subsets of $\mathbb{R}^N$, $\mu$ and $\nu$ be the probability measures on $\mathcal{X}$ and $\mathcal{Y}$, respectively. The Kantorovich formulation of optimal transport is defined by,

$$\mathcal{D}_c(\mu, \nu) := \inf_{\pi \in \Pi(\mu,\nu)} \int_{x,y} c(x,y)\pi(dx,dy),$$

where $\mathcal{D}$ is the OT of $\mu$ into $\nu$ with respect to a cost function $c(\cdot, \cdot)$, and $\Pi(\mu, \nu)$ is a set containing all probability measures $\pi$ whose marginal distributions on $\mathcal{X}$ and $\mathcal{Y}$ correspond to $\mu$ and $\nu$.

Notably, when the cost function is the Euclidean distance between the two arguments, $\mathcal{D}$ is better known as the Wasserstein distance. To facilitate the computation of evaluating OT losses, Cuturi (2013) and Genevay et al. (2017) suggested to add an entropic regularization to $\mathcal{D}_c$. This yields the objective function,

$$\mathcal{D}_{c,\epsilon}(\mu, \nu) = \inf_{\pi \in \Pi(\mu,\nu)} \int_{x,y} \{c(x,y) - \epsilon H(\pi|\mu \otimes \nu)\}\pi(dx,dy),$$

where $H$ denotes the Kullback-Leibler divergence, and $\mu \otimes \nu$ is the product measure of $\mu$ and $\nu$. Such an objective function can be efficiently evaluated by the Sinkhorn algorithm (Cuturi, 2013; Genevay et al., 2017). To alleviate the bias resulting from the entropic regularization term, Cuturi (2013) and Genevay et al. (2017) further considered using the Sinkhorn loss function, defined as, $\widetilde{\mathcal{D}}_{c,\epsilon}(\mu, \nu) = 2\mathcal{D}_{c,\epsilon}(\mu, \nu) - \mathcal{D}_{c,\epsilon}(\mu, \mu) - \mathcal{D}_{c,\epsilon}(\nu, \nu)$.

Next, we present our algorithm for learning the generator $\mathbb{G}_X^{(\ell)}$ in Algorithm 3, where, $\hat{\mathbf{x}}^N$ and $\hat{\mathbf{y}}_\theta^N$ denote the empirical distributions of $\{x_j\}_{j=1}^N$ and $\{y_\theta^j\}_{j=1}^N$, respectively.

---

**Input:** data $\{X_i\}_{i \in \mathcal{I}^{(\ell)}}, \{Z_i\}_{i \in \mathcal{I}^{(\ell)}}$, probability distribution $\zeta$ on the latent space $\mathbb{V}$, initial values of $\theta_0$ and $\varphi_0$, batch size $N$, regularization parameter $\epsilon$, number of Sinkhorn iterations $R$, and learning rate $\alpha$.

**Repeat:**

    *Train discriminator*:

        Sample $\{x^j\}_{j=1}^N$ from $\{(X_i, Z_i)\}_{i \in \mathcal{I}^{(\ell)}}$;

        Sample $\{z^j\}_{j=1}^N$ from $\{Z_i\}_{i \in \mathcal{I}^{(\ell)}}$;

        Sample $\{v^j\}_{j=1}^N$ from $\zeta$;

        $y_\theta^j \leftarrow \{(\mathbb{G}_\theta(z^j, v^j), z^j)\}$ for all $j$;

        $\varphi \leftarrow \varphi + \alpha \nabla_\varphi \widetilde{D}_{c,\epsilon}(\hat{\mathbf{x}}^N, \hat{\mathbf{y}}_\theta^N))$.

    *Train generator*:

        Sample $\{x^j\}_{j=1}^N$ from $\{(X_i, Z_i)\}_{i \in \mathcal{I}^{(\ell)}}$;

        Sample $\{z^j\}_{j=1}^N$ from $\{Z_i\}_{i \in \mathcal{I}^{(\ell)}}$;

        Sample $\{v^j\}_{j=1}^N$ from $\zeta$;

        $y_\theta^j \leftarrow \{(\mathbb{G}_\theta(z^j, v^j), z^j)\}$ for all $j$;

        $\theta \leftarrow \theta - \alpha \nabla_\theta \widetilde{D}_{c,\epsilon}(\hat{\mathbf{x}}^N, \hat{\mathbf{y}}_\theta^N))$.

**until** convergence.

---

**Algorithm 3:** Algorithm for training Sinkhorn GANs by stochastic gradient descent.

## C   ADDITIONAL DETAILS FOR GANS MODEL CHECKING

Let $d_Z$ denote the dimension of $Z$, and $\widehat{\mu}_Z$ the sample average $n^{-1} \sum_i Z_i$. Let $\widetilde{Y}_i = G_Y(Z_i, v_{i,Y})$ denote a simulated sample to approximate the distribution of $Y | Z = Z_i$ obtained by the generator $G_Y$. When $G_Y$ is accurate, we expect the conditional distribution of $\widetilde{Y}_i$ and $Y_i$ given $Z_i$ are similar. As such, for any $d_Z$-dimensional vector $a$, the histograms $\{\widetilde{Y}_i : a^\top(\widetilde{Z}_i - \widehat{\mu}_Z) > 0\}$ and $\{Y_i : a^\top(Z_i - \widehat{\mu}_Z) > 0\}$ should be similar. We sample i.i.d. normal vectors $\{a_g\}_g$ from $N(0, I_{d_Z})$. For each $g$, we plot the histogram $\{Y_i : a_g^\top(Z_i - \widehat{\mu}_Z) > 0\}$ and $\{\widetilde{Y}_i^{(m)} : a_g^\top(Z_i - \widehat{\mu}_Z) > 0\}$. See Figures 3 (a) and (b) for the conditional histograms with two choices of $a_g$, respectively. It can be seen that the Sinkhorn GANs fit the conditional density reasonably well.

The fitted conditional distribution for $P_{X|Z}$ could be checked in a similar fashion as well.

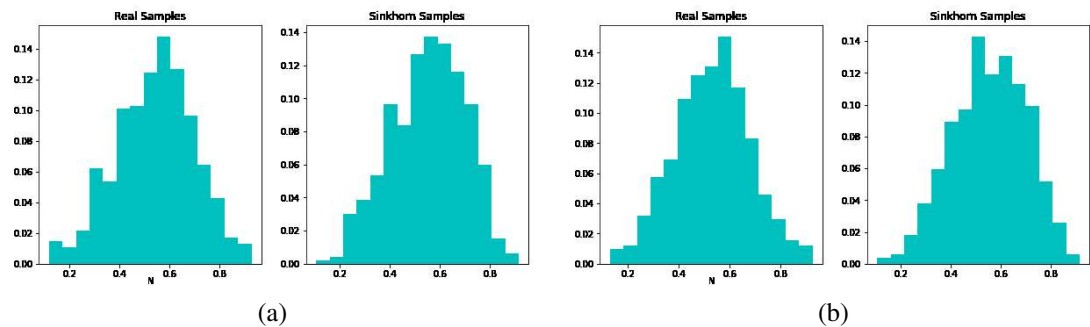

Figure 3: Conditional Histograms. GANs are trained using data generated from the simulation study (see Section 5.1).

We next comment on the performance of the WGANs using Bellot & van der Schaar (2019)'s code. Specifically, we apply their code to their simulation setting as well as our experiments to train the WGAN. We then plot the histograms of the real samples and the WGAN samples scaled between 0 and 1 in Figure 4. It can be seen that the WGANs perform poorly in both settings. For instance,

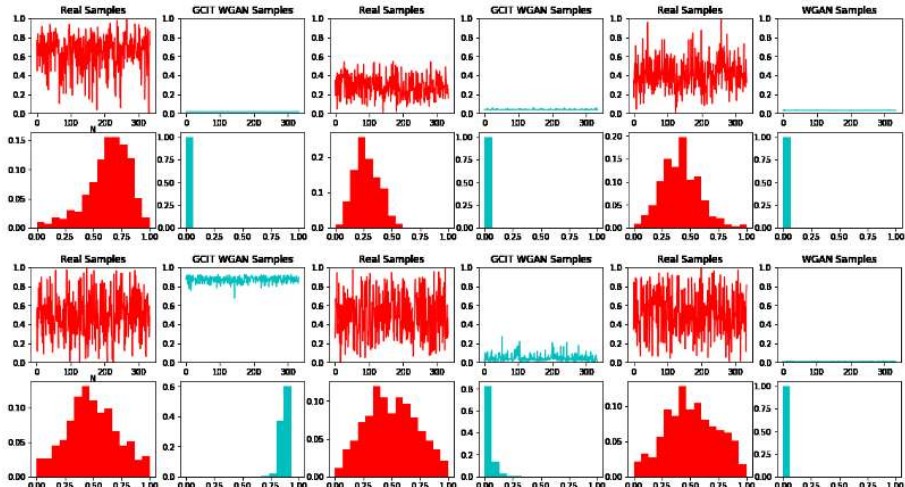

Figure 4: The line plots and histograms of real samples and GCIT WGAN samples (scaled between 0 and 1). The top two panels are the results under Bellot & van der Schaar (2019)'s setup with three different simulations and the bottom top panels, are the results under our setup with three different simulations.

under Bellot & van der Schaar (2019)'s setup, the WGAN samples are very close to zero and do not have much variability. This is very different from the distribution of the real samples.

## D   ADDITIONAL DETAILS FOR MMD

Let $X'$ and $Y'$ be independent copies of $X$ and $Y$ such that they are conditionally independent given $Z$. Note that,

$$\mathrm{E}\{h_1(X')h_2(Y')\} = \mathrm{E}[\mathrm{E}\{h_1(X')|Z\}\mathrm{E}\{h_2(Y')|Z\}],$$

Henceforth,

$$
\begin{aligned}
\phi_{XY} = \mathrm{MMD}(P_{XY}, Q_{XY}|\mathbb{H}_1 \otimes \mathbb{H}_2) &= \sup_{h_1 \in \mathbb{H}_1, h_2 \in \mathbb{H}_2} \Big[\mathrm{E}\{h_1(X)h_2(Y)\} - \mathrm{E}\{h_1(X')h_2(Y')\}\Big] \\
&= \sup_{h_1 \in \mathbb{H}_1, h_2 \in \mathbb{H}_2} \Big[\mathrm{E}\{h_1(X)h_2(Y)\} - \mathrm{E}[\mathrm{E}\{h_1(X)|Z\}\mathrm{E}\{h_2(Y)|Z\}]\Big] \\
&= \sup_{h_1 \in \mathbb{H}_1, h_2 \in \mathbb{H}_2} \Big[\mathrm{E}\{h_1(X)h_2(Y)\} - \mathrm{E}[h_1(X)\mathrm{E}\{h_2(Y)|Z\}] - \mathrm{E}[\{h_1(X)|Z\}h_2(Y)] \\
&\quad + \mathrm{E}[\mathrm{E}\{h_1(X)|Z\}\mathrm{E}\{h_2(Y)|Z\}]\Big] = \sup_{h_1 \in \mathbb{H}_1, h_2 \in \mathbb{H}_2} \mathrm{E}\Big[h_1(X) - E\{h_1(X)|Z\}\Big]\Big[h_2(Y) - E\{h_2(Y)|Z\}\Big].
\end{aligned}
$$

## E   ADDITIONAL NUMERICAL RESULTS

In our simulation settings, we find that DL-CIT, CCIT and KCIT cannot control the type-I error in finite samples. In particular, the type-I errors of DL-CIT are larger than $0.2$ in almost all cases. We first use DL-CIT as an example to investigate its type-I error with a larger sample size. We fix $d_Z = 150$, and generate $Z$ from a standard normal distribution. It can be seen that DL-CIT requires a very large sample size, e.g., $n = 5000$, in order to control the type-I error.

We next conduct additional experiments to investigate the performance of the proposed test with a small sample size, i.e., when $n = 500$. We fix the dimension $d = 100$, generate $Z$ from a standard normal distribution, and set $\delta$ to $0.9$ under $\mathcal{H}_1$. We did not implement DL-CIT and KCIT, because they are no longer valid even when $n = 1000$. Table 2 report the results. It is seen that the proposed test is consistent under this small sample size setting too. Meanwhile, GCIT and RCIT do not have powers under $\mathcal{H}_1$. CCIT has inflated type-I error under $\mathcal{H}_0$. Specifically, its type-I errors under

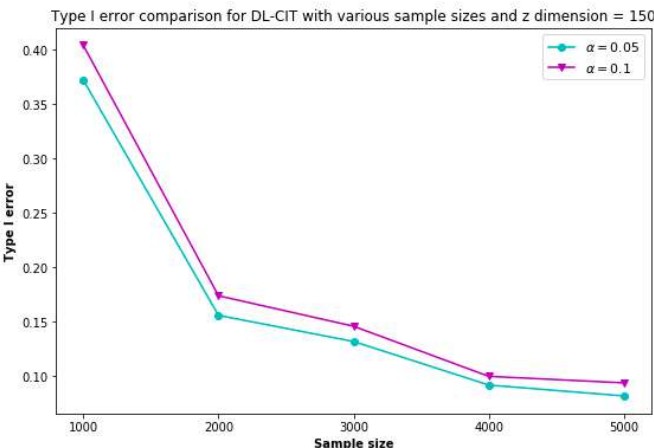

Figure 5: Type-I errors of DL-CIT with larger sample sizes.

| | $\mathcal{H}_0$ | | | | $\mathcal{H}_1$ | | |
| | DGCIT | GCIT | RCIT | CCIT | DGCIT | GCIT | RCIT |
|---|---|---|---|---|---|---|---|
| $\alpha = 0.05$ | 0.04 | 0.06 | 0.05 | 0.08 | 0.22 | 0.07 | 0.07 |
| $\alpha = 0.1$ | 0.06 | 0.13 | 0.11 | 0.13 | 0.35 | 0.10 | 0.10 |

Table 2: Empirical type-I error rate and power of various tests under $\mathcal{H}_0$ and $\mathcal{H}_1$. $n = 500$, $d = 100$, normal $Z$ and $\delta = 0.9$ under $\mathcal{H}_1$.

$\alpha = 0.05$ and 0.1 are 0.08 and 0.13, respectively. The differences $0.08 - 0.05$ and $0.13 - 0.1$ are statistically significant as they exceed the Monte Carlo error $1.96 * \sqrt{0.05 * 0.95/500} = 0.019$ and $1.96\sqrt{0.1 * 0.9/500} = 0.026$. Thus its power becomes meaningless.

## F  PROOFS

We provide the proofs of Proposition 1, and Theorems 2, 3, and 4. We omit the proof of Theorem 1, since it is similar as that of Theorem 2. Theorems 1-4 are established under our choice of the function classes $\mathbb{H}_1$ and $\mathbb{H}_2$, which are set to the classes of neural networks with a single-hidden layer, finitely many hidden nodes, and the sigmoid activation function, as used in our implementation. Meanwhile, the results can be extended to more general cases.

### F.1  PROOF OF PROPOSITION 1

Note that the total variation distance is bounded by 1. Suppose $\mathrm{E}d_{\mathrm{TV}}(\widetilde{P}_{\boldsymbol{X}|\boldsymbol{Z}}, P_{\boldsymbol{X}|\boldsymbol{Z}}) = o(1)$. Then we have $d_{\mathrm{TV}}(\widetilde{P}_{\boldsymbol{X}|\boldsymbol{Z}}, P_{\boldsymbol{X}|\boldsymbol{Z}}) = o_p(1)$. By the dominated convergence theorem, we have $\mathrm{E}d_{\mathrm{TV}}^2(\widetilde{P}_{\boldsymbol{X}|\boldsymbol{Z}}, P_{\boldsymbol{X}|\boldsymbol{Z}}) = o(1)$.

By Theorem 1.2 of Devroye et al. (2018), we have $d_{\mathrm{TV}}(\widetilde{P}_{\boldsymbol{X}|\boldsymbol{Z}}, P_{\boldsymbol{X}|\boldsymbol{Z}})$ is proportional to

$$\min\left[1, \sigma_0^{-1}\sqrt{\sum_{i=1}^{n}\{Z_i^\top(\widehat{\beta} - \beta_0)\}^2}\right].$$

It follows that

$$\frac{1}{\sigma_0}\mathrm{E}\sum_{i=1}^{n}\{Z_i^\top(\widehat{\beta} - \beta_0)\}^2 = o(1).$$

Applying Theorem 1.2 of Devroye et al. (2018) again, we obtain $d_{\mathrm{TV}}(\widetilde{P}_{X|Z=Z_i}, P_{X|Z=Z_i})$ is proportional to

$$\min\left[1, \sigma_0^{-1}|Z_i^\top(\widehat{\beta}-\beta_0)|\right].$$

Thus, we obtain

$$\sum_{i=1}^n \mathrm{E}d_{\mathrm{TV}}^2(\widetilde{P}_{X|Z=Z_i}, P_{X|Z=Z_i}) = o(1).$$

Since the data is exchangeable, we have

$$\mathrm{E}d_{\mathrm{TV}}^2(\widetilde{P}_{X|Z=Z_i}, P_{X|Z=Z_i}) = o(n^{-1}). \tag{6}$$

This shows that when RHS of (2) is $o(1)$, (6) automatically holds.

Next, we show (6) is violated in the linear regression example. By the data exchangability, it suffices to show $\sum_{i=1}^n \mathrm{E}d_{\mathrm{TV}}^2\{\widetilde{P}_{X|Z=Z_i}, P_{X|Z=Z_i}\}$ is not $o(1)$. With some calculations, we obtain

$$\sum_{i=1}^n \mathrm{E}\min\left[1, \sigma_0^{-2}|Z_i^\top(\widehat{\beta}-\beta_0)|^2\right] = \sum_{i=1}^n \mathrm{E}\sigma_0^{-2}|Z_i^\top(\widehat{\beta}-\beta_0)|^2\mathbb{I}\{\sigma_0^{-2}|Z_i^\top(\widehat{\beta}-\beta_0)|^2 \leq 1\}$$

$$+ \sum_{i=1}^n \mathrm{E}\mathbb{I}\{\sigma_0^{-2}|Z_i^\top(\widehat{\beta}-\beta_0)|^2 > 1\} = \sum_{i=1}^n \mathrm{E}\sigma_0^{-2}|Z_i^\top(\widehat{\beta}-\beta_0)|^2 \tag{7}$$

$$- \sum_{i=1}^n \mathrm{E}[\sigma_0^{-2}|Z_i^\top(\widehat{\beta}-\beta_0)|^2 - 1]\mathbb{I}\{\sigma_0^{-2}|Z_i^\top(\widehat{\beta}-\beta_0)|^2 > 1\}.$$

By the definition of $\widehat{\beta}$, we have

$$\sum_{i=1}^n \mathrm{E}\sigma_0^{-2}|Z_i^\top(\widehat{\beta}-\beta_0)|^2 = \frac{1}{\sigma_0^2}\mathrm{E}(\widehat{\beta}-\beta)^\top \boldsymbol{Z}^\top \boldsymbol{Z}(\widehat{\beta}-\beta) = \frac{1}{\sigma_0^2}\mathrm{E}\boldsymbol{\varepsilon}^\top \boldsymbol{Z}(\boldsymbol{Z}^\top \boldsymbol{Z})^{-1}\boldsymbol{Z}^\top\boldsymbol{\varepsilon},$$

where $\boldsymbol{\varepsilon} = (\varepsilon_1, \cdots, \varepsilon_n)^\top$ consist of i.i.d. copies of $\varepsilon$ defined in Example 1. It follows that

$$\sum_{i=1}^n \mathrm{E}\sigma_0^{-2}|Z_i^\top(\widehat{\beta}-\beta_0)|^2 = \frac{1}{\sigma_0^2}\mathrm{E}\boldsymbol{\varepsilon}^\top \boldsymbol{Z}(\boldsymbol{Z}^\top \boldsymbol{Z})^\top \boldsymbol{Z}^\top\boldsymbol{\varepsilon} = \frac{1}{\sigma_0^2}\mathrm{trace}\left\{\mathrm{E}\boldsymbol{\varepsilon}\boldsymbol{\varepsilon}^\top \boldsymbol{Z}(\boldsymbol{Z}^\top \boldsymbol{Z})^{-1}\boldsymbol{Z}^\top\right\}$$

$$= \mathrm{trace}\left\{\mathrm{E}\boldsymbol{Z}(\boldsymbol{Z}^\top \boldsymbol{Z})^{-1}\boldsymbol{Z}^\top\right\} = d_Z, \tag{8}$$

where $d_Z$ is the dimension of $Z$.

In the following, we show

$$\sum_{i=1}^n \mathrm{E}\sigma_0^{-2}|Z_i^\top(\widehat{\beta}-\beta_0)|^2\mathbb{I}\{\sigma_0^{-2}|Z_i^\top(\widehat{\beta}-\beta_0)|^2 \geq 1\} = o(1). \tag{9}$$

Combining this together with (7) and (8) yields

$$\sum_{i=1}^n \mathrm{E}\min\left[1, \sigma_0^{-2}|Z_i^\top(\widehat{\beta}-\beta_0)|^2\right] \geq d_Z - o(1) \geq 1 - o(1),$$

and hence $\sum_{i=1}^n \mathrm{E}d_{\mathrm{TV}}^2\{\widetilde{P}_{X|Z=Z_i}, Q_X^{(n)}(\cdot|Z_i)\} \geq 1 - o(1)$. The proof is hence completed.

It remains to show (9), or equivalently,

$$\mathrm{E}n\sigma_0^{-2}|Z_i^\top(\widehat{\beta}-\beta_0)|^2\mathbb{I}\{\sigma_0^{-2}|Z_i^\top(\widehat{\beta}-\beta_0)|^2 \geq 1\} = o(1).$$

We have already shown that $\mathrm{E}n\sigma_0^{-2}|Z_i^\top(\widehat{\beta}-\beta_0)|^2 = d_Z$. By dominated convergence theorem, it suffices to show

$$n\sigma_0^{-2}|Z_i^\top(\widehat{\beta}-\beta_0)|^2\mathbb{I}\{\sigma_0^{-2}|Z_i^\top(\widehat{\beta}-\beta_0)|^2 \geq 1\} = o_p(1).$$

By definition, it suffices to show

$$\Pr\left(\sigma_0^{-2}|Z_i^\top(\widehat{\beta}-\beta_0)|^2 \geq 1\right) \to 0.$$

However, this is immediate to see by Markov's inequality as

$$\mathrm{E}\sigma_0^{-2}|Z_i^\top(\widehat{\beta}-\beta_0)|^2 = \frac{d_Z}{n} \to 0.$$

This completes the proof of Proposition 1.

### F.2 PROOF OF THEOREM 2

We begin by providing an upper bound for the function classes $\mathbb{H}_1$ and $\mathbb{H}_2$. Recall that $\theta_{1,b}$ and $\theta_{2,b}$ are generated according to $N(0, 2I_{d_1}/d_1)$ and $N(0, 2I_{d_2}/d_2)$, respectively. Since $\mathbb{H}_1$ and $\mathbb{H}_2$ are classes of neural networks with a single hidden layer and finitely many hidden nodes. Both $d_1$ and $d_2$ are finite. All the elements $\{\sqrt{d_1}\theta_{1,b,j}\}_{b,j}$ and $\{\sqrt{d_2}\theta_{2,b,j}\}_{b,j}$ are i.i.d. standard normal. For any standard normal variable $\mathbb{Z}$, we have for any $t \geq 1$ that

$$
\Pr(|\mathbb{Z}| > t) \leq 2 \int_t^{+\infty} \phi(z)dz \leq 2 \int_t^{+\infty} z\phi(z)dz = 2\exp\left(-\frac{t^2}{2}\right),
$$

where $\phi(\cdot)$ denotes the standard normal density function. Setting $t = c^*\sqrt{2\log n}$ for some constant $c^* > 0$, we obtain $\Pr(|\mathbb{Z}| > t) \leq n^{-c^*}$. It follows from Bonferroni's inequality that

$$
\Pr\left(\left\{\max_{b,j}|\sqrt{d_1}\theta_{1,b,j}| > t\right\} \bigcap \left\{\max_{b,j}|\sqrt{d_2}\theta_{2,b,j}| > t\right\}\right) \leq Bd_1d_2n^{-c^*}. \tag{10}
$$

Since $d_1$ and $d_2$ are finite and $B = O(n^c)$ for some constant $c > 0$, the right-hand-side converges to zero with proper as long as $c^* > c$. Consequently, all the weights can be uniformly bounded by $O(\sqrt{\log n})$. Since the sigmoid function is set to the activation function, the function classes are bounded by $O(\sqrt{\log n})$ with probability tending to 1. Without loss of generality, we assume this event holds throughout the proofs of Theorems 2–4.

Define a test statistic

$$
T^{**} = \max_{b_1,b_2}\widehat{\sigma}_{b_1,b_2}^{-1}\left|\frac{1}{n}\sum_{i=1}^n\left[h_{1,b_1}(X_i) - \frac{1}{M}\sum_{m=1}^M h_{1,b_1}(X_i^{(m)})\right]\left[h_{2,b_2}(Y_i) - \frac{1}{M}\sum_{m=1}^M h_{2,b_2}(Y_i^{(m)})\right]\right|,
$$

where the $\widehat{\sigma}_{b_1,b_2}$ is constructed based on $\{\widetilde{X}_i^{(m)}\}_m$ and $\{\widetilde{Y}_i^{(m)}\}_m$, instead of $\{X_i^{(m)}\}_m$ and $\{Y_i^{(m)}\}_m$. Thus, it suffices to show $|T - T^{**}| = O_p(n^{-2\kappa})$ and $|T^* - T^{**}| = O_p(n^{-2\kappa})$.

Consider the difference $|T - T^{**}|$. For any sequences $\{a_n\}_n, \{b_n\}_n$, we have

$$
|\max_n|a_n| - \max_n|b_n|| \leq \max_n|a_n - b_n|. \tag{11}
$$

Consequently, the difference $|T - T^{**}|$ is upper bounded by $I_1 + I_2 + I_3$ where

$$
I_1 = \max_{b_1,b_2}\widehat{\sigma}_{b_1,b_2}^{-1}\left|\frac{1}{n}\sum_{i=1}^n\left[\frac{1}{M}\sum_{m=1}^M\{h_{1,b_1}(X_i^{(m)}) - h_{1,b_1}(\widetilde{X}_i^{(m)})\}\right]\left[h_{2,b_2}(Y_i) - \frac{1}{M}\sum_{m=1}^M h_{2,b_2}(Y_i^{(m)})\right]\right|,
$$

$$
I_2 = \max_{b_1,b_2}\widehat{\sigma}_{b_1,b_2}^{-1}\left|\frac{1}{n}\sum_{i=1}^n\left[h_{1,b_1}(X_i) - \frac{1}{M}\sum_{m=1}^M h_{1,b_1}(X_i^{(m)})\right]\left[\frac{1}{M}\sum_{m=1}^M\{h_{2,b_2}(Y_i^{(m)}) - h_{2,b_2}(\widetilde{Y}_i^{(m)})\}\right]\right|,
$$

$$
I_3 = \max_{b_1,b_2}\widehat{\sigma}_{b_1,b_2}^{-1}\left|\frac{1}{n}\sum_{i=1}^n\left[\frac{1}{M}\sum_{m=1}^M\{h_{1,b_1}(X_i^{(m)}) - h_{1,b_1}(\widetilde{X}_i^{(m)})\}\right]\left[\frac{1}{M}\sum_{m=1}^M\{h_{2,b_2}(Y_i^{(m)}) - h_{2,b_2}(\widetilde{Y}_i^{(m)})\}\right]\right|.
$$

By definition, we have $\min\widehat{\sigma}_{b_1,b_2} \geq \sqrt{\epsilon}$ for some constant $\epsilon > 0$. It suffices to show $I_j^* = O_p(n^{-2\kappa}\log n)$ for $j = 1, 2, 3$ where

$$
I_1^* = \max_{b_1,b_2}\left|\frac{1}{n}\sum_{i=1}^n\left[\frac{1}{M}\sum_{m=1}^M\{h_{1,b_1}(X_i^{(m)}) - h_{1,b_1}(\widetilde{X}_i^{(m)})\}\right]\left[h_{2,b_2}(Y_i) - \frac{1}{M}\sum_{m=1}^M h_{2,b_2}(Y_i^{(m)})\right]\right|,
$$

$$
I_2^* = \max_{b_1,b_2}\left|\frac{1}{n}\sum_{i=1}^n\left[h_{1,b_1}(X_i) - \frac{1}{M}\sum_{m=1}^M h_{1,b_1}(X_i^{(m)})\right]\left[\frac{1}{M}\sum_{m=1}^M\{h_{2,b_2}(Y_i^{(m)}) - h_{2,b_2}(\widetilde{Y}_i^{(m)})\}\right]\right|,
$$

$$
I_3^* = \max_{b_1,b_2}\left|\frac{1}{n}\sum_{i=1}^n\left[\frac{1}{M}\sum_{m=1}^M\{h_{1,b_1}(X_i^{(m)}) - h_{1,b_1}(\widetilde{X}_i^{(m)})\}\right]\left[\frac{1}{M}\sum_{m=1}^M\{h_{2,b_2}(Y_i^{(m)}) - h_{2,b_2}(\widetilde{Y}_i^{(m)})\}\right]\right|.
$$

The number of folds $L$ is finite, as such, it suffices to show $I_j^{(\ell)} = O_p(n^{-2\kappa} \log n)$ for $j = 1, 2, 3$ and $\ell = 1, \cdots, L$, where

$$I_1^{(\ell)} = \max_{b_1,b_2} \left| \frac{1}{n} \sum_{i \in \mathcal{I}^{(\ell)}} \left[ \frac{1}{M} \sum_{m=1}^{M} \{h_{1,b_1}(X_i^{(m)}) - h_{1,b_1}(\widetilde{X}_i^{(m)})\} \right] \left[ h_{2,b_2}(Y_i) - \frac{1}{M} \sum_{m=1}^{M} h_{2,b_2}(Y_i^{(m)}) \right] \right|,$$

$$I_2^{(\ell)} = \max_{b_1,b_2} \left| \frac{1}{n} \sum_{i \in \mathcal{I}^{(\ell)}} \left[ h_{1,b_1}(X_i) - \frac{1}{M} \sum_{m=1}^{M} h_{1,b_1}(X_i^{(m)}) \right] \left[ \frac{1}{M} \sum_{m=1}^{M} \{h_{2,b_2}(Y_i^{(m)}) - h_{2,b_2}(\widetilde{Y}_i^{(m)})\} \right] \right|,$$

$$I_3^{(\ell)} = \max_{b_1,b_2} \left| \frac{1}{n} \sum_{i \in \mathcal{I}^{(\ell)}} \left[ \frac{1}{M} \sum_{m=1}^{M} \{h_{1,b_1}(X_i^{(m)}) - h_{1,b_1}(\widetilde{X}_i^{(m)})\} \right] \left[ \frac{1}{M} \sum_{m=1}^{M} \{h_{2,b_2}(Y_i^{(m)}) - h_{2,b_2}(\widetilde{Y}_i^{(m)})\} \right] \right|.$$

The rest of the proof is divided into four steps. In the first three steps, we show $I_j^{(\ell)} = O_p(n^{-2\kappa} \log n)$ for $j = 1, 2, 3$. n the last step, we show $T^* - T^{**} = O_p(n^{-2\kappa} \log n)$.

**Step 1.** Without loss of generality, suppose functions in $\mathbb{H}_1$ and $\mathbb{H}_2$ are bounded by $\log n$ in absolute values. By Bernstein's inequality, we have

$$\Pr\left\{ \left| \sum_{m=1}^{M} h_{1,b}(X_i^{(m)}) - M\mathrm{E}\{h_{1,b}(X_i)|Z_i\} \right| \geq t \right\} \leq 2 \exp\left\{ -\frac{t^2}{2(M \log n + t\sqrt{\log n}/3)} \right\},$$

for any $b$ and $i$. Set $t = \sqrt{3(c+2)M} \log n$ where the constant $c$ is defined in the statement of Theorem 1. For sufficiently large $n$, we have $t\sqrt{\log n}/3 \leq M \log n/2$. It follows that

$$\Pr\left\{ \left| \sum_{m=1}^{M} h_{1,b}(X_i^{(m)}) - M\mathrm{E}\{h_{1,b}(X_i)|Z_i\} \right| \geq \sqrt{3(c+2)M} \log n \right\} \leq \frac{2}{n^{c+2}}.$$

By Bonferroni's inequality, we obtain

$$\Pr\left\{ \max_{b \in \{1,\cdots,B\}} \max_{i \in \{1,\cdots,n\}} \left| \sum_{m=1}^{M} h_{1,b}(X_i^{(m)}) - M\mathrm{E}\{h_{1,b}(X_i)|Z_i\} \right| \geq \sqrt{3(c+2)M} \log n \right\}$$

$$\leq Bn \max_{b \in \{1,\cdots,B\}} \max_{i \in \{1,\cdots,n\}} \Pr\left\{ \left| \sum_{m=1}^{M} h_{1,b}(X_i^{(m)}) - M\mathrm{E}\{h_{1,b}(X_i)|Z_i\} \right| \geq \sqrt{3(c+2)M} \log n \right\} \leq \frac{2Bn}{n^{c+2}}.$$

Under the condition $B = O(n^c)$, we obtain with probability $1 - O(n^{-1})$ that

$$\max_{b \in \{1,\cdots,B\}} \max_{i \in \{1,\cdots,n\}} \left| \sum_{m=1}^{M} h_{1,b}(X_i^{(m)}) - M\mathrm{E}\{h_{1,b}(X_i)|Z_i\} \right| \leq O(1)n^{-1/2} \log n, \tag{12}$$

as $M$ is proportional to $n$, and $O(1)$ denotes some positive constant.

Similarly, we can show

$$\max_{b \in \{1,\cdots,B\}} \max_{i \in \mathcal{I}^{(\ell)}} \left| \sum_{m=1}^{M} h_{1,b}(\widetilde{X}_i^{(m)}) - M \int_x h_{1,b}(x) \widetilde{P}_{X|Z=Z_i}^{(\ell)}(dx) \right| \leq O(1)\sqrt{n} \log n,$$

with probability $1 - O(n^{-1})$, as well. Combining this together with (12), we obtain with probability $1 - O(n^{-1})$ that

$$\max_{\substack{b \in \{1,\cdots,B\} \\ i \in \mathcal{I}^{(\ell)}}} \left| \sum_{m=1}^{M} \{h_{1,b}(X_i^{(m)}) - h_{1,b}(\widetilde{X}_i^{(m)})\} - M \int_x h_{1,b}(x)\{P_{X|Z=Z_i}(dx) - \widetilde{P}_{X|Z=Z_i}^{(\ell)}(dx)\} \right| \tag{13}$$

$$\leq O(1)\sqrt{n} \log n.$$

Conditional on $Z_i$, the expectation of $h_{2,b_2}(Y_i) - M^{-1} \sum_{m=1}^{M} h_{2,b_2}(Y_i^{(m)})$ equals zero. Under the null hypothesis, the expectation of $M^{-1} \sum_{m=1}^{M} \{h_{1,b_1}(X_i^{(m)}) - h_{1,b_1}(\widetilde{X}_i^{(m)})\}\{h_{2,b_2}(Y_i) -$

$M^{-1} \sum_{m=1}^{M} h_{2,b_2}(Y_i^{(m)})\}$ equals zero as well. Applying Bernstein's inequality again, we can similarly show with probability tending to 1 that $I_1^{(\ell)}$ can be upper bounded by

$$O(1)(\sigma n^{-1/2} \log^{3/2} n + n^{-1} \log^2 n), \tag{14}$$

where $O(1)$ denotes some positive constant and

$$\sigma^2 = \max_{b_1,b_2} E \left| \frac{1}{M} \sum_{m=1}^{M} \{h_{1,b_1}(X_i^{(m)}) - h_{1,b_1}(\widetilde{X}_i^{(m)})\}\{h_{2,b_2}(Y_i) - \frac{1}{M} \sum_{m=1}^{M} h_{2,b_2}(Y_i^{(m)})\} \right|^2$$

$$\leq \max_{b_1} E \left| \frac{1}{M} \sum_{m=1}^{M} \{h_{1,b_1}(X_i^{(m)}) - h_{1,b_1}(\widetilde{X}_i^{(m)})\} \right|^2 \log n.$$

Let $\mathcal{A}$ denote the event in (13). The last term on the second line can be bounded from above by

$$\max_{b_1,i} E \left| \frac{1}{M} \sum_{m=1}^{M} \{h_{1,b_1}(X_i^{(m)}) - h_{1,b_1}(\widetilde{X}_i^{(m)})\} \right|^2 \mathbb{I}(\mathcal{A}) \log n \tag{15}$$

$$+ \max_{b_1,i} E \left| \frac{1}{M} \sum_{m=1}^{M} \{h_{1,b_1}(X_i^{(m)}) - h_{1,b_1}(\widetilde{X}_i^{(m)})\} \right|^2 \mathbb{I}(\mathcal{A}^c) \log n. \tag{16}$$

Since $M$ is proportional to $n$, by (11), the term on the first line of (15) is upper bounded by

$$O(1) \left\{ n^{-1} \log^2 n + \max_{\substack{b \in \{1,\cdots,B\} \\ i \in \mathcal{I}^{(\ell)}}} E \left| \int_x h_{1,b}(x) \{\widetilde{P}_{X|Z=Z_i}^{(\ell)}(dx) - P_{X|Z=Z_i}(dx)\} \right|^2 \right\} \log n.$$

By the boundedness of the function class $\mathbb{H}_1$, it can be further bounded from above by

$$O(1) \left\{ n^{-1} \log^3 n + E d_{\mathrm{TV}}^2(\widetilde{P}_{X|Z}^{(\ell)}, P_{X|Z}) \log^2 n \right\}. \tag{17}$$

Under the current conditions, the above quantity is of the order $O(n^{-2\kappa} \log^2 n)$. Consequently, (15) is or the order $O(n^{-2\kappa} \log^2 n)$.

Note that the event $\mathcal{A}$ occurs with probability at least $1 - O(n^{-1})$. By the boundedness of the function class $\mathbb{H}_1$, (16) is of the order $O(n^{-1} \log^2 n)$.

To summarize, we have further that $\sigma^2$ is of the order $O(n^{-2\kappa} \log^2 n)$. This implies $\mathcal{I}_1^{(\ell)}$ can be bounded from above by $O(n^{-1/2-\kappa} \log^{5/2} n)$. This yields $\mathcal{I}_1^{(\ell)} = O_p(n^{-2\kappa} \log n)$.

**Step 2.** Step 2 can be proven in a similar manner as Step 1 and is thus omitted.

**Step 3.** Under $H_0$, the expectation of

$$\frac{1}{|\mathcal{I}^{(\ell)}|} \sum_{i \in \mathcal{I}^{(\ell)}} \left[ \frac{1}{M} \sum_{m=1}^{M} \{h_{1,b_1}(X_i^{(m)}) - h_{1,b_1}(\widetilde{X}_i^{(m)})\} \right] \left[ \frac{1}{M} \sum_{m=1}^{M} \{h_{2,b_2}(Y_i^{(m)}) - h_{2,b_2}(\widetilde{Y}_i^{(m)})\} \right]$$

equals

$$E \int_x h_{1,b_1}(x) \{\widetilde{P}_{X|Z}^{(\ell)}(dx) - P_{X|Z}(dx)\} \int_y h_{2,b_2}(y) \{\widetilde{P}_{Y|Z}^{(\ell)}(dy) - P_{Y|Z}(dy)\}.$$

Similar to (17), its absolute value can be upper bounded by

$$E d_{\mathrm{TV}} \{\widetilde{P}_{X|Z=Z_i}^{(\ell)}, P_{X|Z}\} d_{\mathrm{TV}} \{\widetilde{P}_{Y|Z=Z_i}^{(\ell)}, P_{Y|Z}\} \log n.$$

Under the given conditions, it follows from Cauchy-Schwarz inequality that

$$E d_{\mathrm{TV}} \{\widetilde{P}_{X|Z=Z_i}^{(\ell)}, P_{X|Z}\} d_{\mathrm{TV}} \{\widetilde{P}_{Y|Z=Z_i}^{(\ell)}, P_{Y|Z}\}$$

$$\leq \frac{1}{2} E d_{\mathrm{TV}}^2 \{\widetilde{P}_{X|Z=Z_i}^{(\ell)}, P_{X|Z}\} + \frac{1}{2} E d_{\mathrm{TV}}^2 \{\widetilde{P}_{Y|Z=Z_i}^{(\ell)}, P_{Y|Z}\} = O(n^{-2\kappa}).$$

This yields that

$$\max_{b_1,b_2}\left|\mathrm{E}\int_x h_{1,b_1}(x)\{\widetilde{P}_{X|Z}^{(\ell)}(dx)-P_{X|Z}(dx)\}\int_y h_{2,b_2}(y)\{\widetilde{P}_{Y|Z}^{(\ell)}(dy)-P_{Y|Z}(dy)\}\right|=O(n^{-2\kappa}\log n).$$

Using similar arguments in Step 1, we can show that

$$I_3^{(\ell)}-\max_{b_1,b_2}\left|\mathrm{E}\int_x h_{1,b_1}(x)\{\widetilde{P}_{X|Z}^{(\ell)}(dx)-P_{X|Z}(dx)\}\int_y h_{2,b_2}(y)\{\widetilde{P}_{Y|Z}^{(\ell)}(dy)-P_{Y|Z}(dy)\}\right|=O_p(n^{-2\kappa}\log n).$$

Thus, we obtain $I_3^{(\ell)}=O_p(n^{-2\kappa}\log n)$. This completes the proof of Step 3.

**Step 4.** Denote by $\widehat{\sigma}_{b_1,b_2}^{*2}$ the variance estimator with $\{\widetilde{X}_i^{(m)}\}_m$ and $\{\widetilde{Y}_i^{(m)}\}_m$ replaced by $\{X_i^{(m)}\}_m$ and $\{Y_i^{(m)}\}_m$. Using (11), the difference between $T^*$ and $T^{**}$ is upper bounded by

$$\max_{b_1,b_2}|\widehat{\sigma}_{b_1,b_2}^{-1}-\widehat{\sigma}_{b_1,b_2}^{*-1}|\left|n^{-1}\sum_{i=1}^n\left[h_{1,b_1}(X_i)-\frac{1}{M}\sum_{m=1}^M h_{1,b_1}(X_i^{(m)})\right]\left[h_{2,b_2}(Y_i)-\frac{1}{M}\sum_{m=1}^M h_{2,b_2}(Y_i^{(m)})\right]\right|.$$

Under $H_0$, similar to (13), we can show

$$\max_{b_1,b_2}\left|\left|n^{-1}\sum_{i=1}^n\left[h_{1,b_1}(X_i)-\frac{1}{M}\sum_{m=1}^M h_{1,b_1}(X_i^{(m)})\right]\left[h_{2,b_2}(Y_i)-\frac{1}{M}\sum_{m=1}^M h_{2,b_2}(Y_i^{(m)})\right]\right|\right|$$
$$=O_p(n^{-1/2}\log^{3/2} n).$$

To show $|T^*-T^{**}|=O_p(n^{-2\kappa})$, it suffices to show $\max_{b_1,b_2}|\widehat{\sigma}_{b_1,b_2}^{-1}-\widehat{\sigma}_{b_1,b_2}^{*-1}|=O_p(n^{-\bar{c}})$ for some constant $\bar{c}>0$. Since both $\widehat{\sigma}_{b_1,b_2}^{-1}$ and $\widehat{\sigma}_{b_1,b_2}$ are well-bounded away from zero. It suffices to show $\max_{b_1,b_2}|\widehat{\sigma}_{b_1,b_2}^2-\widehat{\sigma}_{b_1,b_2}^{*2}|=O_p(n^{-\bar{c}})$.

Using similar arguments in Steps 1 and 3, we can show

$$\max_{b_1,b_2}\left|\widehat{\sigma}_{b_1,b_2}^2-\frac{n}{n-1}\mathrm{Var}\Big([h_{1,b_1}(X)-\mathrm{E}\{h_{1,b_1}(X)|Z\}][h_{2,b_2}(Y)-\mathrm{E}\{h_{2,b_2}(Y)|Z\}]\Big)\right|=O_p(n^{-\bar{c}}),$$

and

$$\max_{b_1,b_2}\left|\widehat{\sigma}_{b_1,b_2}^{*2}-\frac{n}{n-1}\mathrm{Var}\Big([h_{1,b_1}(X)-\mathrm{E}\{h_{1,b_1}(X)|Z\}][h_{2,b_2}(Y)-\mathrm{E}\{h_{2,b_2}(Y)|Z\}]\Big)\right|=O_p(n^{-\bar{c}}).$$

This completes the proof of Theorem 2.

### F.3 Proof of Theorem 3

In the proof of Theorem 2, we have shown $T-T^*=O_p(n^{-2\kappa}\log n)$. Using similar arguments in Step 4 of the proof of Theorem 2, we can show $T^*-T^{***}=O_p(n^{-2\kappa}\log n)$ where

$$T^{***}=\max_{b_1,b_2}\sigma_{b_1,b_2}^{-1}\left|n^{-1}\sum_{i=1}^n\left[h_{1,b_1}(X_i)-\frac{1}{M}\sum_{m=1}^M h_{1,b_1}(X_i^{(m)})\right]\left[h_{2,b_2}(Y_i)-\frac{1}{M}\sum_{m=1}^M h_{2,b_2}(Y_i^{(m)})\right]\right|,$$

where

$$\sigma_{b_1,b_2}^2=\max\left\{\frac{n}{n-1}\mathrm{Var}\Big([h_{1,b_1}(X)-\mathrm{E}\{h_{1,b_1}(X)|Z\}][h_{2,b_2}(Y)-\mathrm{E}\{h_{2,b_2}(Y)|Z\}]\Big),\epsilon\right\}.$$

By (12), using similar arguments in $I_1$ in the proof of Theorem 2, we can show $T^{***}-T^{****}=O_p(n^{-2\kappa}\log n)$ where

$$T^{****}=\max_{b_1,b_2}\sigma_{b_1,b_2}^{-1}\left|n^{-1}\sum_{i=1}^n[h_{1,b_1}(X_i)-\mathrm{E}\{h_{1,b_1}(X_i)|Z_i\}]\left[h_{2,b_2}(Y_i)-\frac{1}{M}\sum_{m=1}^M h_{2,b_2}(Y_i^{(m)})\right]\right|.$$

Similarly, we can show $T^{****} - T_0 = O_p(n^{-2\kappa} \log n)$ where

$$T_0 = \max_{b_1, b_2} \sigma_{b_1, b_2}^{-1} \left| n^{-1} \sum_{i=1}^{n} [h_{1,b_1}(X_i) - \mathrm{E}\{h_{1,b_1}(X_i)|Z_i\}] [h_{2,b_2}(Y_i) - \mathrm{E}\{h_{2,b_2}(Y_i)|Z_i\}] \right|.$$

To summarize, we have shown $T - T_0 = O_p(n^{-2\kappa} \log n)$. Since we require $\kappa > 1/4$, we obtain

$$\sqrt{n}(T - T_0) = o_p(\log^{-1/2} n). \tag{18}$$

Define a $B^2 \times B^2$ matrix $\Sigma_0$ whose $\{b_1 + B(b_2 - 1), b_3 + B(b_4 - 1)\}$th entry is given by

$$\mathrm{Cov} \left( \sigma_{b_1, b_2}^{-1} [h_{1,b_1}(X_i) - \mathrm{E}\{h_{1,b_1}(X_i)|Z_i\}] [h_{2,b_2}(Y_i) - \mathrm{E}\{h_{2,b_2}(Y_i)|Z_i\}], \right.$$

$$\left. \sigma_{b_3, b_4}^{-1} [h_{1,b_3}(X_i) - \mathrm{E}\{h_{1,b_3}(X_i)|Z_i\}] [h_{2,b_4}(Y_i) - \mathrm{E}\{h_{2,b_4}(Y_i)|Z_i\}] \right).$$

In the following, we show

$$\sup_t |\mathrm{Pr}(\sqrt{n} T_0 \le t|\mathcal{H}_0) - \mathrm{Pr}(\|N(0, \Sigma_0)\|_\infty \le t)| = o(1). \tag{19}$$

When $B$ is finite, this is implied by the classical weak convergence results. When $B$ diverges with $n$, we require $B = O(n^c)$ for some constant $c > 0$. By the definition of $\sigma_{b_1, b_2}$, the variance of

$$\sigma_{b_1, b_2}^{-1} [h_{1,b_1}(X_i) - \mathrm{E}\{h_{1,b_1}(X_i)|Z_i\}] [h_{2,b_2}(Y_i) - \mathrm{E}\{h_{2,b_2}(Y_i)|Z_i\}]$$

is bounded from above by $(n - 1)/n$. Moreover, combining the boundedness assumption on the function spaces $\mathbb{H}_1$ and $\mathbb{H}_2$ together with the definition of $\sigma_{b_1, b_2}$ yields that

$$\left\{ \sigma_{b_1, b_2}^{-1} [h_{1,b_1}(X_i) - \mathrm{E}\{h_{1,b_1}(X_i)|Z_i\}] [h_{2,b_2}(Y_i) - \mathrm{E}\{h_{2,b_2}(Y_i)|Z_i\}] : b_1, b_2 \in \{1, \cdots, B\} \right\}$$

are uniformly bounded away from infinity. Similar to Corollary 4.1 of Chernozhukov et al. (2014), we can show that (19) holds. Suppose $B = 1$. Then $N(0, \Sigma_0)$ is a single normal variable. This implies that

$$\sigma_{b_1, b_2}^{-1} n^{-1/2} \sum_{i=1}^{n} [h_{1,b_1}(X_i) - \mathrm{E}\{h_{1,b_1}(X_i)|Z_i\}] [h_{2,b_2}(Y_i) - \mathrm{E}\{h_{2,b_2}(Y_i)|Z_i\}]$$

is asymptotically normal with zero mean.

Combining (19) together with (18) yields

$$\mathrm{Pr}(\sqrt{n} T \le t|\mathcal{H}_0) \ge \mathrm{Pr}(\|N(0, \Sigma_0)\|_\infty \le t - \epsilon_0 \log^{-1/2} n) - o(1),$$
$$\mathrm{Pr}(\sqrt{n} T \le t|\mathcal{H}_0) \le \mathrm{Pr}(\|N(0, \Sigma_0)\|_\infty \le t + \epsilon_0 \log^{-1/2} n) + o(1), \tag{20}$$

for any sufficiently small $\epsilon_0 > 0$, where the little-o terms are uniform in $t$.

Using similar arguments in Step 4 and Step 5 of the proof of Theorem 2, we can show that $\|\widehat{\Sigma} - \Sigma_0\|_{\infty, \infty} = O_p(n^{-\bar{c}})$ for some constant $\bar{c} > 0$. Using similar arguments for (20) and also Lemma 3.1 of Chernozhukov et al. (2015), we have that

$$\mathrm{Pr}(\sqrt{n} T \le t|\mathcal{H}_0) \ge \mathrm{Pr}(\|N(0, \widehat{\Sigma})\|_\infty \le t - 2\epsilon_0 \log^{-1/2} n|\widehat{\Sigma}) - o(1),$$
$$\mathrm{Pr}(\sqrt{n} T \le t|\mathcal{H}_0) \le \mathrm{Pr}(\|N(0, \widehat{\Sigma})\|_\infty \le t + 2\epsilon_0 \log^{-1/2} n|\widehat{\Sigma}) + o(1),$$

for any sufficiently small $\epsilon_0 > 0$. Since the little-o terms are uniform in $t \in \mathbb{R}$, we obtain

$$\sup_t |\mathrm{Pr}(\sqrt{n} T \le t|\mathcal{H}_0) - \mathrm{Pr}(\|N(0, \widehat{\Sigma})\|_\infty \le t|\widehat{\Sigma})| \le o(1)$$

$$+ \sup_t |\mathrm{Pr}(\|N(0, \widehat{\Sigma})\|_\infty \le t + 2\epsilon \log^{-1/2} n|\widehat{\Sigma}) - \mathrm{Pr}(\|N(0, \widehat{\Sigma})\|_\infty \le t - 2\epsilon \log^{-1/2} n|\widehat{\Sigma})|.$$

The term on the second line can be bounded by $O(1)\epsilon \log^{1/2} B \log^{-1/2} n$ where $O(1)$ denotes some positive constant, by Theorem 1 of Chernozhukov et al. (2017). Since $B = O(n^c)$, $\log^{1/2} B \log^{-1/2} n = O(1)$. As $\epsilon$ grows to zero, this term becomes negligible. Consequently, we obtain

$$\sup_t |\mathrm{Pr}(\sqrt{n} T \le t|\mathcal{H}_0) - \mathrm{Pr}(\|N(0, \widehat{\Sigma})\|_\infty \le t|\widehat{\Sigma})| \le o(1).$$

As such, the distribution of our test statistic can be well-approximated by that of the bootstrap samples. This completes the proof of Theorem 3.

### F.4 PROOF OF THEOREM 4

According to the universal approximation theorem (Barron, 1993), neural networks with a single hidden layer and the sigmoid activation function are universal approximators. Under $\mathcal{H}_1^*$, there exist two neural networks functions $f(X)$ and $g(Y)$, with a single hidden layer and the sigmoid activation function, such that

$$\mathrm{E}[f(X) - \mathrm{E}\{f(X)|Z\}][g(Y) - \mathrm{E}\{g(Y)|Z\}] \neq 0.$$

Note that $f$ ($g$) can be represented by linear combinations of functions in $\mathbb{H}_1$ ($\mathbb{H}_2$). It implies that there exists some constant $c^* > 0$ such that $\mathrm{GCM}^*(h_1^*(X), h_2^*(Y)) > c^*$ for some $h_1^* \in \mathbb{H}_1$ and $h_2^* \in \mathbb{H}_2$. Let $\theta_1^*$ and $\theta_2^*$ be the corresponding parameters such that $h_1^* = h_{1,\theta_1^*}$, $h_2^* = h_{2,\theta_2^*}$.

We next show that $\mathrm{GCM}^*(h_{1,\theta_1}(X), h_{2,\theta_2}(Y))$ is a Lipschitz continuous function of $(\theta_1, \theta_2)$. Note that $h_{1,\theta_1}(X)$ and $h_{2,\theta_2}(Y)$ are Lipschitz continuous functions of $\theta_1$ and $\theta_2$, respectively. For any $\theta_{1,1}, \theta_{1,2} \in \mathbb{R}^{d_1}, \theta_{2,1}, \theta_{2,2} \in \mathbb{R}^{d_2}$, we have

$$|\mathrm{GCM}^*(h_{1,\theta_1}(X), h_{2,\theta_2}(Y)) - \mathrm{GCM}^*(h_{1,\theta_1}(X), h_{2,\theta_2}(Y))|$$
$$\leq |\mathrm{E}[h_{1,1}(X) - \mathrm{E}\{h_{1,1}(X)|Z\} - h_{1,2}(X) + \mathrm{E}\{h_{2,1}(X)|Z\}][h_{2,1}(Y) - \mathrm{E}\{h_{2,1}(Y)|Z\}]| \quad (21)$$
$$+ |\mathrm{E}[h_{1,2}(X) - \mathrm{E}\{h_{1,2}(X)|Z\}][h_{2,1}(Y) - \mathrm{E}\{h_{2,1}(Y)|Z\} - h_{2,2}(Y) + \mathrm{E}\{h_{2,2}(Y)|Z\}]|. \quad (22)$$

Since $\mathbb{H}_2$ is bounded function class, RHS of (21) is bounded from above by

$$O(1)\mathrm{E}|h_{1,1}(X) - \mathrm{E}\{h_{1,1}(X)|Z\} - h_{1,2}(X) + \mathrm{E}\{h_{2,1}(X)|Z\}|\sqrt{\log n},$$

where $O(1)$ denotes some positive constant. By Jensen's inequality, the above quantity can be further bounded from above by

$$O(1)\mathrm{E}|h_{1,1}(X) - h_{1,2}(X)|2\sqrt{\log n} \leq L\|\theta_{1,1} - \theta_{1,2}\|_2\sqrt{\log n},$$

for some constant $L > 0$. Using similar arguments, we can show RHS of (22) is bounded from above by $L\|\theta_{2,1} - \theta_{2,2}\|_2$. To summarize, we have shown that

$$|\mathrm{GCM}^*(h_{1,\theta_1}(X), h_{2,\theta_2}(Y)) - \mathrm{GCM}^*(h_{1,\theta_1}(X), h_{2,\theta_2}(Y))|$$
$$\leq L(\|\theta_{1,1} - \theta_{1,2}\|_2 + \|\theta_{2,1} - \theta_{2,2}\|_2)\sqrt{\log n}.$$

As such, for any sufficiently small $\epsilon > 0$, there exists a neighborhood $\mathcal{N} = \{(\theta_1, \theta_2) : \|\theta_j - \theta_j^*\|_2 \leq \delta \log^{-1/2} n\}$ for some constant $\delta > 0$ around $(\theta_1^*, \theta_2^*)$ such that $\mathrm{GCM}^*(h_{1,\theta_1}(X), h_{2,\theta_2}(Y)) \geq \epsilon$ for any $(\theta_1, \theta_2)$ that belongs to this neighborhood.

Since we use multivariate normal to generate $(\theta_{1,b}, \theta_{2,b})$ and the dimensions $d_1$ and $d_2$ are finite, the probability that $(\theta_{1,b}, \theta_{2,b})$ belongs to this neighborhood is strictly greater than $O(\log^{-c_1} n)$ for some constant $c_1 > 0$. Since $B = c_0 n^c$, the probability that at least one pair of parameters $(\theta_{1,b_1}, \theta_{2,b_2})$ belongs to this neighborhood approaches one. Consequently, we have

$$\max_{b_1, b_2} \mathrm{GCM}^*(h_{1,b_1}(X), h_{2,b_2}(Y)) \geq \epsilon,$$

with probability tending to 1.

Using similar arguments in the proof of Theorems 2 and 3, we can show that $|T - \max_{b_1,b_2} \mathrm{GCM}^*(h_{1,b_1}(X), h_{2,b_2}(Y))| = o_p(1)$ and $\widetilde{T}_j = o_p(1)$. Consequently, both probabilities $\Pr(T < \epsilon/2)$ and $\Pr(\widetilde{T}_j \geq \epsilon/2)$ converge to zero. As such, the probability that the p-value is greater than $\alpha$ is bounded by the probability that $\Pr(T < \epsilon/2)$, and hence converges to zero. This completes the proof of Theorem 4.

