# OpenReview forum: "Double Generative Adversarial Networks for Conditional Independence Testing"
_ICLR.cc/2021/Conference — Reject_

### Official Review · AnonReviewer4 · 2020-10-19
**Sophisticated conditional independence test with well-studied theoretical guarantees.**

**Rating:** 6
**Confidence:** 4

**Review:**

The authors propose a non-parametric conditional independence test that approximates distances in a Hilbert space of functions using a generative approach, both evaluating conditional expectations using samples from GANs and evaluating a supremum over a set of functions previously generated at random. The test incorporates benefits from different lines of research and is demonstrated to outperform alternatives in well-known benchmarks. I think this study is interesting and serves as a good example of the synergies that may be achieved by combining powerful function approximators and strong statistical arguments.

My biggest concern is that conditional independence testing is an unsupervised problem, there is therefore little scope to test for the goodness of fit of hyperparameter choices or the accuracy of approximations. The proposed approach has plenty of user-defined parameters yet the sensitivity of performance to different choices is not investigated nor is there a discussion of sensible values to be recommended in practice.

Some specific questions on this thread are as follows.
- Appendix A says: "The performance of GANs is largely affected by the regularization parameter and the number of Sinkhorn iterations R.", should we expect performance to vary a lot in practice?
- Appendix C tests the goodness of fit of the GAN approximation by comparing observed and estimated distributions p(y|x). Why should this be a good indication that GANs approximate well p(y|z) or p(x|z)?

Most consistency guarantees are asymptotic in nature. With increasing conditioning set size, the fact that complex conditional distributions need to be approximated and that separate sets of samples are needed to train and compute the test statistic, I worry that a very large number of samples will be needed to achieve good performance. Would experiments as a function of sample size be possible to include in the paper?


Minor comments:
- The test itself is quite involved, with many moving parts that are described over various pages in the paper. I would recommend to have a higher-level summary of the approach perhaps using Figure 2 in the Appendix earlier in the paper.
- A single data generating mechanism is used for performance comparisons. I think different multiple different choices here would be needed to test different aspects of the model.

---

> ### Author Response · Authors · 2020-11-24
> **Response to Reviewer 4**
>
> We greatly appreciate your valuable comments, which have helped lead to a much-improved manuscript. We have uploaded the revised manuscript that take into account all your suggestions. In the following, we present our point-by-point responses to your questions.
>
> **Choice of the hyperparameter**.
> Thank you very much for this thoughtful comment. As discussed in Section A, line 4-9, the performance of our procedure is not sensitive to the choice of tuning parameters such as $M$ and $L$, and we recommend to set $B$ to 30-50 to achieve a good balance between computational cost and power.
>
> Other parameters, such as the number of regularization parameter $\epsilon$, the number of Sinkhorn iterations $R$, and the number of hidden layers and nodes in the neural networks might have an impact on the quality of the generator and thus affect the testing results. In practice, we suggest to tune these parameters by investigating the goodness-of-fit of the resulting generator. This could be achieved by comparing the conditional histogram of the generated samples to that of the true samples. Please see our response to your Point #3 for more details. As we have shown in Theorems 3 and 4, our test is consistent as long as the fitted GANs achieve a certain rate of convergence.
>
> We have added the related discussions in the paper (Appendix A, page 11, line 14-16).
>
> **The regularization parameter and the number of Sinkhorn iterations**.
> The quality of GANs could vary with difference choices of these parameters. As in our response to your point #1, we suggest to tune these parameters by checking the conditional histogram of the resulting generator.
>
> **Conditional histogram**.
> We apologize for the typo in our previous manuscript. We shall compare the observed and the estimated distribution $P_{Y|Z}$ instead of $P_{Y|X}$.
>
> We next explain why the conditional histogram can be used for model checking. Let $\widetilde{Y_i}=G_Y(Z_i, v_{i,Y})$ denote a simulated sample to approximate the distribution of $Y|Z=Z_i$ obtained by the generator $G_Y$. When $G_Y$ is accurate, we expect the conditional distributions of $\widetilde{Y}_i$ and $Y_i$ given $Z_i$ are similar. As such, for any $d_Z$-dimensional vector $a$, the histograms $\{\widetilde{Y}_i: a^\top (Z_i-\widehat{\mu}_Z)>0\}$ and $\{Y_i: a^\top (Z_i-\widehat{\mu}_Z)>0\}$ shall be similar. Consequently, we can randomly draw multiple $a$ and plot those histograms to check the goodness-of-fit of the generator.
> We have added the related discussions in the paper (Appendix C, page 12).
>
> **Experiments with small sample size**.
> Following your suggestion, we have conducted additional experiments to investigate the performance of our test under a small sample size.  Due to the limited time, we fix $n=500$, $d=100$, generate $Z$ from a standard normal distribution, and set $b$ to $0.9$ under $\mathcal{H}_1$. We did not implement DL-CIT and KCIT as they are not valid even when $n=1000$. Results are reported in Table 2 (page 14).
>
> It is seen that the proposed test is consistent under this setting. GCIT and RCIT do not have powers under $\mathcal{H}_1$. CCIT has inflated type-I errors under $\mathcal{H}_0$. Specifically, its type-I errors under $\alpha=0.05$ and $0.1$ are $0.08$ and $0.13$, respectively. The differences $0.08-0.05$ and $0.13-0.1$ are statistically significant as they exceed the Monte Carlo error $1.96\times\sqrt{0.05\times 0.95/500}=0.019$ and $1.96\times \sqrt{0.1\times 0.9/500}=0.026$, where we use 500 simulation replications. Its power thus becomes meaningless.
>
> We plan to add more scenarios with $n=500$ should the paper be accepted.
>
> **Some minor comments**.
> We have moved Figure 2 to the main text (page 4). Due to the time constraint, we did not implement other data generating mechanisms. We plan to address this comment should the paper be accepted.

---

### Official Review · AnonReviewer2 · 2020-10-26
**Official Blind Review #1**

**Rating:** 6
**Confidence:** 3

**Review:**

This paper considers the problem of conditional independence testing, especially when the variables are high-dimensional. The authors proposed a double GAN based algorithm. Two GANs are designed to learn the conditional probability distributions P_{X|Z} and P_{Y|Z}, then used to generate samples to compute the test statistic. It is proved that the error of the test statistic is O_p(n^{-2k} \log n) when the total variation error of the GANS is O(n^{-k}). So to ensure that the test statistic converges, only o(\log^{-1/2} n) rate is required for the total variation error of the GANs.

The result of this paper is quite strong. Compared to the paper of Bellot & van der Schaar (2019)  which requires the TV error of GAN to be o(n^{-1/2}), this paper significantly reduce the requirement to o(\log^{-1/2} n).  I did not check the full proof.

Questions to the authors: In the description of Bellot & van der Schaar(2019), it seems that the data used for training the GAN and the data used for computing testing statistic are shared. However, in the proposed algorithm (Algorithm 1), the data are split into L blocks where GANs are trained by L-1 blocks of data and test statistics are computed by the other. Is this a critical reason for the improvement of convergence rate? If so, how will the convergence rate be if we apply data splitting to Bellot & van der Schaar (2019)? If not, what is the key reason for the improvement of convergence rate, double-GAN or the randomly generated h functions?

---

> ### Author Response · Authors · 2020-11-24
> **Response to Reviewer 2**
>
> We greatly appreciate your valuable comments, which have helped lead to a much-improved manuscript. We have uploaded the revised manuscript that take into account all your suggestions. In the following, we present our point-by-point responses to your questions.
>
> **Improvement of the convergence rate**. We first clarify the requirement of the total variation error. We have reduced the requirement from $o(n^{1/2})$ to $o(n^{1/4}\log^{1/2} n)$, as we require the test statistic $T$ to converge to the oracle test $T^*$ (defined in Section 4, page 6, line 13) at a rate of $o(n^{-1/2})$ (see the discussion below Theorem 2). Note that the oracle test converges at a rate of $O_p(n^{-1/2})$. When this condition is achieved, our test is asymptotically equivalent to the oracle test and is valid. The relaxed condition is achievable (see Examples 3-5 on page 6), whereas the original $o(n^{1/2})$ convergence rate requirement is likely to be violated (see Example 1 on page 2).
>
> We next clarify that the proposed test relaxes the condition by estimating two generators (double-GANs) and making the test statistic doubly-robust (Theorem 1, page 6).  The notion of “doubly-robustness” property comes from the classical semiparametric theory in statistics (see e.g., Tsiatis, 2007, Semiparametric theory and missing data). Specifically, a doubly robust procedure applies both types of models simultaneously, and produces a consistent estimate if either of the two models has been consistently estimated.
>
> In our setup, the double-robustness property guarantees our test statistic is consistent to the oracle test statistic, as long as either the estimated conditional distribution for $P_{X|Z}$ or $P_{Y|Z}$ is consistent. In addition, the difference between our test and the oracle test decays to zero at a faster rate than the convergence rate of the estimated conditional distribution. In contrast, the GCIT test statistic converges at the same rate as the estimated conditional distribution. As such, our procedure requires a much weaker condition. We have added the related discussion in the paper (page 4 line 6-9 and page 6, line 17-22).
>
> Finally, we comment on the use of sample splitting. Sample splitting cannot improve the rate of convergence. It helps further reduce the type-I error of the test (see page 5, the first paragraph), and is commonly used for statistical inference (see e.g., Chernozhukov et al., 2018, Double/debiased machine learning for treatment and structural parameters).

---

### Official Review · AnonReviewer3 · 2020-10-28
**Novel approach combine GAN technique with Conditional Independence Testing; still room to improve**

**Rating:** 5
**Confidence:** 4

**Review:**


The paper proposed a novel simulation based testing procedure for conditional independence: X\perp Y | Z. The testing procedure incorporate the techniques of GAN, which is especially useful for dealing with high-dimensional data. The testing procedure first learn the generative adversarial network that is able to simulate the conditional distribution of X|Z and Y|Z and check a particular kernel-based independence criteria presented in Eq.(4). Instead of kernel-based method that takes the supreme over a class of RKHS functions, the proposed testing procedure searches the maximum "discrepancy" over a class of neural network function by simulation. Empirical results show a well controlled type-I error and better test power performances compare to existing methods and a cancer data application is discussed.

This is an interesting paper combining kernel testing and GAN setting which can potentially broaden the scope of kernel-based tests. In addition to the contributions, there are some unclear parts in presentation and concerns on the proposed schemes.

1. The kernel-based test, e.g. MMD relies on the positive definiteness of the RKHS. The notion of characteristic kernel ensures that the null hypothesis hold iff the test statistics is 0. In this work, do you or how do you ensure the positive definiteness? By taking the max over an absolute value in Eq.(5) does ensure non-negativity of the test statistics, but not yet the characteristic notion. Instead of RKHS kernel, it may end up being a Krein space kernel, i.e. RKKS. This may reduce the test power of the statistics in particular scenario, but it is unclear.

2. From the flow chart in Fig.2, the two generators and discriminators are trained separately. How does the notion of "doubly" comes in. If the training is interactive between Y|Z and X|Z network, how is it done? This part is not entirely clear.

3. Is the notion of robustness come from the learned conditional density (or generator) is robust against noisy samples or some form of adversarial attack, or some other notion? Not entirely clear just from Thereom 1, instead, it sound more like asymptotic property.

4. The number of "bootstrap" functions B need to go to infinity, maybe it will be better to emphasize that.


5. Building up from the GCM type of statistic, instead of MMD, it may be better to refer Hilbert Space Independence Criterion (HSIC) instead of MMD.
Appendix D derivation, despite mathematically correct,  need more steps: the add-in and subtract h_1(X) times conditional expectation of h_2(Y)|Z term, otherwise it does not supply additional information to the main text.

6. Experiment findings: in top panel of Fig.1, there is a spike on dim=150 for DL-CIT, is there any intuition why this happens? And KCIT and CCIT papers both claimed their controlled type-I error, what is the reason for this scenario that they fail?

Thanks for the presentation.

---

> ### Author Response · Authors · 2020-11-24
> **Response to Reviewer 3 Part 2**
>
> 6. KCIT and CCIT control type I error **asymptotically**, but there is no guarantee for finite samples. Our simulations indicate a better finite sample control of type I error for our proposed method. We next comment on the inflated type-I errors of some of the baseline methods, including DL-CIT, KCIT and CCIT.
>
>     In Appendix E, we use DL-CIT as an example to investigate its type-I error under a larger sample size. We fix d=150. In general, the type I-error decays at $n$ increases. We find DL-CLT requires a very large sample size (e.g., $n=5000$) in order to achieve the nominal type-I error (see Page 14, Figure 5).
>
>     Finally, we clarify that we study the high-dimensional conditional independence testing problem in our paper. The number of covariates is allowed to increase along with the sample size. Thanks to the double-robustness property of our procedure and the capacity of deep neural nets in estimating the conditional distributions in high dimensions, the proposed test is theoretically valid. In contrast, the theoretical guarantees for many other tests are established under the scenario where the dimension of the conditional variables is fixed. It remains unknown whether those tests are valid in high dimensions.

---

> ### Author Response · Authors · 2020-11-24
> **Response to Reviewer 3 Part 1**
>
> We greatly appreciate your valuable comments, which have helped lead to a much-improved manuscript. We have uploaded the revised manuscript that take into account all your suggestions. In the following, we present our point-by-point responses to your questions.
>
> 1. The MMD criterion is defined as
>    $MMD_{\mathbb{F}}(P,Q)) = \sup_{f\in \mathbb{F}} \left( E_{W_1\sim P}f(W_1) - E_{W_2\sim Q}f(W_2)  \right)$
> where $\mathbb{F}$ is a space of bounded functions.
> The space $\mathbb{F}$ determines the power properties of the test based on MMD, and the bigger $\mathbb{F}$ is, the more alternatives the test has power for. Our test can be used in principle with any $\mathbb{F}$, and in the paper, we have used a class of neural networks, which can approximate any function (Barron, 1993, Universal approximation bounds for superpositions of a sigmoidal function). Henceforth, we obtain the power for any fixed alternative.
>
>     Alternatively, if $\mathbb{F}$ is the unit ball in a reproducing kernel Hilbert space (RKHS) based on a characteristic kernel, then $MMD_{\mathbb{F}}(P,Q) \ge 0$ with equality if and only if $P=Q$. This choice of $\mathbb{F}$ would lead to another implementation of our test, and could naturally be investigated in future. However, the paper is not about the choice of $\mathbb{F}$, rather, about the estimation of the conditional marginal distributions.
>
>     Next, we remark that there exists non-RKHS classes that satisfy the characteristic notion as well.  Examples of non-RKHS classes of functions for which $MMD_{\mathbb{F}}(P,Q)\ge 0$ with equality if and only if $P=Q$ are given in Section 7.2 of Gretton et al., 2012, A kernel two-sample test.
>
>     Regarding Krein spaces, for any Krein space, there is a Hilbert space consisting of the same set of functions (Langer, 2001, Krein space). So Krein spaces do not have any added benefit or drawback over Hilbert spaces for defining MMD and play no role in our paper.
>
> 2. The notion of “doubly” comes from the fact that we learn **two** generators, one to approximate the conditional distribution X|Z and the other to approximate Y|Z. This shares a similar spirit as the “double machine learning” method (Chernozhukov et al., 2018, Double/debiased machine learning for treatment and structural parameters) for causal inference, where two machine learning methods are used to construct the estimator for the average treatment effects.
>
> 3. The notion of “doubly-robustness” property comes from the classical semiparametric theory in statistics (see e.g., Tsiatis, 2007). Specifically, a doubly robust procedure applies both types of models simultaneously, and produces a consistent estimate if either of the two models has been consistently estimated.
>
>     In our proposal, we learn two generators to approximate the conditional distributions X|Z and Y|Z. Theorem 1 shows our test statistic is doubly robust. Specifically, it converges to the oracle test statistic as long as one of the generators is consistent. It also implies the convergence rate of our test statistic is faster than the estimated conditional distribution (Theorem 2). In contrast, the GCIT test statistic converges at the same rate as the estimated conditional distribution. As such, our procedure requires a much weaker condition.
>
>     We have added the related discussions in the paper (see page 4, line 6-9 and page 6, line 17-22).
>
> 4. Thanks for the suggestion. We have emphasized that B shall diverge to infinity in the paper (page 4, line-11).
>
> 5. The reviewer asks whether HSIC might be used rather than MMD. We think there is a misunderstanding here. HSIC can be viewed as an application of MMD, comparing a joint bivariate distribution $P_{XY}$ with the product of its marginals $P_XP_Y$. Our test is yet another application of MMD, and we compare the bivariate distribution $P_{XY}$ with $Q_{XY}$, where $Q_{XY}$ is the marginal distribution of $Q_{XYZ}=P_{X|Z}P_{Y|Z}P_Z$. Whereas for HSIC is MMD with $\mathbb{F}$ being an RKHS, we take $\mathbb{F}$ to be a class of neural networks in our paper.
>
>     Thanks for pointing out the issue in the Appendix D derivation, those steps have been added (page 13).

---

### Official Review · AnonReviewer1 · 2020-10-29
**Seems sound, but some unclear points.**

**Rating:** 5
**Confidence:** 2

**Review:**

- Overview

This paper develops a test for conditional independence using GAN.
Conditional independence is one of the well-known problems, and calculating the conditional distribution of a random variable is a challenge.
The authors pointed out in Proposition 1 that the existing method, named GCIT, cannot avoid a non-negligible approximation bias.
The authors newly developed test combines the conditional distribution with GAN and regression-based and MMD-based tests to construct a valid test under weaker conditional requirements than the previous method.

- Comments.

The paper points out some important issues with existing research.
However, there are a few things I don't understand.

The experiments show that the GCIT appear to have no power at all, but the experiments of the original paper (Bellot and van der Schaar (2019)) report that GCIT has sufficient power.
Where does this discrepancy come from?
The submitted paper says that it uses a similar setting to Bellot and van der Schaar (2019), so I would like to know why the results are so different.

The theoretical advantages and their relationship to the experiment are less clear.
The paper states that the conditions required for GCIT are unsatisfied in general, but does this show up in the experimental results?
The Type I errors appear to be a bit more or less dominant, but not by much.
Could that be the reason for the poor performance of GCIT in the analysis for power?
If so, then it should be clear why there is a significant difference with Bellot and van der Schaar (2019), as discussed above.

I didn't understand why the conditions by the submitted paper are weaker.
I don't doubt the accuracy, but if the technical points are not properly explained, it is not kind for readers.
I want a clear explanation that can weaken the conditions.

How is the computational time of the proposed method?
To be a practical method, the computation time should be short, but if we use the GAN, it would be a big cost.
Of course, this is true for all studies using GANs, not only this paper, but also for all studies using GANs, but it is important for practical purposes, so please let me know.

---

> ### Author Response · Authors · 2020-11-24
> **Response to Reviewer 1**
>
> We greatly appreciate your valuable comments, which have helped lead to a much-improved manuscript. We have uploaded the revised manuscript that take into account all your suggestions. In the following, we present our point-by-point responses to your questions.
>
> **The discrepancy between GCIT’s performance in Bellot and van der Schaar (2019)’s and our experiments.**
> First, we note that, although their results show GCIT achieves a large power in the experiments, these results may not be completely right. In particular, we used their code to check the quality of their fitted GANs and found these GANs performed very poorly (see Figure 4 in Appendix C, page 13). We thus suspect their code only works under their particular setting, and might not be applicable to more general settings. In our numerical experiments, instead of using their fitted WGANs, we use the Sinkhorn GAN to implement GCIT.
>
> Second, we describe the difference between the settings used in Bellot and van der Schaar (2019) and our paper. In Bellot and van der Schaar (2019), they considered the following setting to test $H_1$,
>  \begin{eqnarray*}
> 	X=f(a_f^\top Z+\epsilon_f)\qquad    \textrm{and}\qquad   Y=g(a_g^\top Z + b X + \epsilon_g),
>   \end{eqnarray*}
> where the transformation functions f and g are randomly chosen among a set of functions. In our paper, we choose two particular transformation functions, by setting f and g to be sine and cosine functions. This makes the alternative hypothesis much harder to detect, so that the differences in powers between different testing methods are relatively distinctive. We suspect this explains the failure of GCIT under our setup.
>
> **The theoretical advantage and their relationship to the experiment.**
> In the paper, we have shown that GCIT requires a strong condition to control the type-I error, and this condition is in general not satisfied. We remark that this does not necessarily mean GCIT would have an inflated type-I error for **every** setting. In our experiment, GCIT does have inflated type I errors in some cases. In particular, when $Z$ is normal, $d_Z$ = 250, and $\alpha = 0.1$, its empirical size is close to $0.15$. The difference $0.15-0.1$ is not much, but it is **statistically significant** as it exceeds the Monte Carlo error $1.96\times \sqrt{0.1\times 0.9/500}=0.0263$, when we use $500$ replications to produce the results.
>
> In Appendix E (page 14), we conduct additional simulations to investigate the performance of various tests with a small sample size. It can be seen from Table 2 that GCIT has inflated type-I errors as well. The empirical rejection probability 0.13 is **significantly** larger than the nominal level 0.1.
>
> We remark that for any testing procedure, its power generally increases with its type-I error. Although GCIT does not show a large type-I error in all cases, it suffers from a low power.
>
> **Explanation on the weaker condition.**
> The proposed procedure relaxes the condition by making the test statistic doubly-robust (see Theorem 1, page 6).  The notion of “doubly-robustness” property comes from the classical semiparametric theory in statistics (see e.g., Tsiatis, 2007). Specifically, a doubly robust procedure applies both types of models simultaneously, and produces a consistent estimate if either of the two models has been consistently estimated.
>
> In our paper, we introduce an “oracle” test statistic in Section 4 (page 6, line 9) that works as well as if the conditional distributions X|Z and Y|Z were known. The oracle test is valid as it does not require to estimate the conditional distributions.  The double-robustness property guarantees our test statistic is consistent to the oracle test statistic as long as either the estimated conditional distribution for $P_{X|Z}$ or $P_{Y|Z}$ is consistent. In addition, the difference between our test and the oracle test decays to zero at a faster rate than the convergence rate of the estimated conditional distribution (Theorem 2, page 6). In contrast, the GCIT test statistic converges at the same rate as the estimated conditional distribution. As such, our procedure requires a much weaker condition.
>
> We have added the related discussions in the paper (page 4 line 6-9 and page 6, line 17-22).
>
> **Computation time**.
> We discussed the implementation and the time complexity of our test in Section 5.1 (page 7). Specifically, it took about 2.5 minutes to compute our test. In comparison, it took 2 minutes to compute CCIT, and 20 seconds to compute GCIT and DL-CIT. We have reported the running times of these baseline methods in Appendix A (page 11).

---

### Decision · Program_Chairs · 2021-01-07
**Final Decision**

**Decision:**

Reject

**Comment:**

This paper discusses the conditional independence test using GAN.   In the same way as GCIT (Bellot & van der Schaar, 2019), they realize sampling under the null hypothesis by generating sample from P(X|Z) approximately with GAN.  They propose to use a test statistic defined by the maximum of generalized covariance measures (GCM) over random neural networks.  They theoretically discuss the advantage of GCM and show the asymptotic results of the proposed test statistic, which demonstrates improved justification over GCIT.  Experimental results show favorable performances over existing conditional independence tests.

The proposed method gives an advance in the methodology of conditional independence tests for continuous domain, which is an important but difficult problem because of the difficulty of obtaining the null distribution.  In the line of Bellot & van der Shcaar (2019), they solve it using the strong conditional sampling ability of GAN, which is an important research area. The theoretical analysis and experimental results are also making good contributions.

However, there are some weakness in the proposed method and comparison with existing methods. First, as R4 points out, there are many hyperparameters in the proposed method, and their choice is not easy.  While the authors addressed some aspects of this issue in their rebuttal and revision, it is still unclear how to justify the choice of B, the functions h_j, and the neural networks of GAN, which should potentially have significant influence on the test performance.  Second, the comparison with Bellot and van der Schaar (2019) is not very clear.  In the paper, the GCIT has been used with the distance correlation, which is known to be an instance of HSIC (MMD) with a specific choice of positive definite kernel (Sejdinovic et al 2013).  The HSIC can be formulated as the maximum of generalized covariance measures over the unit ball of the RKHS.  Thus, the difference of GCIT with distance correlation and the proposed methods are essentially the difference of the function classes for the maximum.  On the other hand, the experimental results show significant difference in the test performance.  I think more elaborate and careful comparison is needed for these two methods.

Overall, the paper is a good contribution on the topic.  However, the evaluation of the reviewers is not high enough to justify the acceptance in the high competition of ICLR.  I encourage the authors complete their work by reflecting reviewers’ comments and submit this work to another conference or journal.

Reference:
Sejdinovic, D., Sriperumbudur, B., Gretton, A., & Fukumizu, K. (2013). Equivalence of distance-based and RKHS-based statistics in hypothesis testing. Annals of Statistics, 41(5), 2263–2291.